# AI-SARAH: Adaptive and Implicit Stochastic Recursive Gradient Methods

**Zheng Shi**  *Shi.Zheng.TFLS@gmail.com*
*IBM*
*United States of America*

**Abdurakhmon Sadiev**  *abdurakhmon.sadiev@kaust.edu.sa*
*King Abdullah University of Science and Technology (KAUST)*
*Thuwal*
*Saudi Arabia*

**Nicolas Loizou**  *nloizou@jhu.edu*
*Johns Hopkins University*
*Baltimore*
*United States of America*

**Peter Richtárik**  *peter.richtarik@kaust.edu.sa*
*King Abdullah University of Science and Technology (KAUST)*
*Thuwal*
*Saudi Arabia*

**Martin Takáč**  *Takac.MT@gmail.com*
*Mohamed bin Zayed University of Artificial Intelligence (MBZUAI)*
*Masdar City, Abu Dhabi*
*United Arab Emirates (UAE)*

**Reviewed on OpenReview:** *https://openreview.net/forum?id=WoXJFsJ6Zw*

## Abstract

We present *AI-SARAH*, a practical variant of *SARAH*. As a variant of *SARAH*, this algorithm employs the stochastic recursive gradient yet adjusts step-size based on local geometry. *AI-SARAH* implicitly computes step-size and efficiently estimates local Lipschitz smoothness of stochastic functions. It is fully adaptive, tune-free, straightforward to implement, and computationally efficient. We provide technical insight and intuitive illustrations on its design and convergence. We conduct extensive empirical analysis and demonstrate its strong performance compared with its classical counterparts and other state-of-the-art first-order methods in solving convex machine learning problems.

## 1 Introduction

We consider the unconstrained finite-sum optimization problem

$$\min_{w \in \mathcal{R}^d} \left[ P(w) \stackrel{\text{def}}{=} \frac{1}{n} \sum_{i=1}^{n} f_i(w) \right]. \tag{1}$$

This problem is prevalent in machine learning tasks where $w$ corresponds to the model parameters, $f_i(w)$ represents the loss on the training point $i$, and the goal is to minimize the average loss $P(w)$ across the training points. In machine learning applications, (1) is often considered the loss function of Empirical Risk Minimization (ERM) problems. For instance, given a classification or regression problem, $f_i$ can be defined as

logistic regression or least square by $(x_i, y_i)$ where $x_i$ is a feature representation and $y_i$ is a label. Throughout the paper, we assume that each function $f_i$, $i \in [n] \stackrel{\text{def}}{=} \{1, ..., n\}$, is smooth and convex, and there exists an optimal solution $w^*$ of (1).

## 1.1 Main Contributions

We propose **A**daptive and **I**mplicit **S**toch**A**stic **R**ecursive Gr**A**dient Algorit**H**m (*AI-SARAH*), a practical variant of stochastic recursive gradient methods (Nguyen et al., 2017) to solve (1). This practical algorithm explores and adapts to local geometry. It is adaptive at full scale yet requires zero effort of tuning hyper-parameters. The extensive numerical experiments demonstrate that our tune-free and fully adaptive algorithm is capable of delivering a consistently competitive performance on various datasets, when comparing with *SARAH*, *SARAH+* and other state-of-the-art first-order methods, all equipped with fine-tuned hyper-parameters (which are selected from $\approx 5,000$ runs for each problem). This work provides a foundation on studying adaptivity (of stochastic recursive gradient methods) and demonstrates that a **truly adaptive stochastic recursive algorithm can be developed in practice.**

## 1.2 Related Work

Stochastic gradient descent (*SGD*) (Robbins & Monro, 1951; Nemirovski & Yudin, 1983; Shalev-Shwartz et al., 2007; Nemirovski et al., 2009; Gower et al., 2019) is the workhorse for training supervised machine learning problems that have the generic form (1).

In its generic form, *SGD* defines the new iterate by subtracting a multiple of a stochastic gradient $g(w_t)$ from the current iterate $w_t$. That is,

$$w_{t+1} = w_t - \alpha_t g(w_t).$$

In most algorithms, $g(w)$ is an unbiased estimator of the gradient (i.e., a stochastic gradient), $\mathbb{E}[g(w)] = \nabla P(w), \forall w \in \mathcal{R}^d$. However, in several algorithms (including the ones from this paper), $g(w)$ could be a biased estimator, and convergence guarantees can still be well obtained.

**Adaptive step-size selection.** The main parameter to guarantee the convergence of *SGD* is the *step-size*. In recent years, several ways of selecting the step-size have been proposed. For example, an analysis of *SGD* with constant step-size ($\alpha_t = \alpha$) or decreasing step-size has been proposed in Moulines & Bach (2011); Ghadimi & Lan (2013); Needell et al. (2016); Nguyen et al. (2018); Bottou et al. (2018); Gower et al. (2019; 2021) under different assumptions on the properties of (1).

More recently, *adaptive / parameter-free* methods (Duchi et al., 2011; Kingma & Ba, 2015; Bengio, 2015; Li & Orabona, 2018; Vaswani et al., 2019; Liu et al., 2019a; Ward et al., 2019; Loizou et al., 2021) that adapt the step-size as the algorithms progress have become popular and are particularly beneficial when training deep neural networks. Normally, in these algorithms, the step-size does not depend on parameters that might be unknown in practical scenarios, like the smoothness or the strongly convex parameter.

In Section 4.1, we explain how the update rule of the proposed *AI-SARAH* (Algorithm 2) involves solving a specific sub-problem for selecting the optimal step-size in the current iteration. This idea is novel, but it is closely related to the recent work of Loizou et al. (2021) on the Stochastic Polyak Step-size (SPS) for SGD, where the optimal choice, to some extent, of step-size is selected in each iteration of SGD. Loizou et al. (2021) provided several convergence guarantees of SGD with SPS, including linear convergence to a neighborhood for solving strongly convex problems and sublinear rate of convergence to a neighborhood for convex and non-convex problems. It was also shown that SPS is particularly effective in step-size selections under the interpolation setting, which enables SGD to converge to the true solution at a fast rate matching the deterministic case. The results of Loizou et al. (2021) were later extended to different settings. Gower et al. (2021) provided analysis of SGD with SPS for structured non-convex problems while D'Orazio et al. (2021) proved convergence of stochastic mirror descent under the mirror stochastic Polyak stepsize (mSPS). More recently, Orvieto et al. proposed Decreasing SPS (DecSPS) as a step-size selection for SGD, a novel modification of SPS, which guarantees convergence (sublinear) to the exact minimizer - without a priori

knowledge of the problem's parameters. Li et al. (2023) recently extended the SPS to include curvature information.

**Random vector $g(w_t)$ and variance reduced methods.** One of the most remarkable algorithmic breakthroughs in recent years was the development of variance-reduced stochastic gradient algorithms for solving finite-sum optimization problems. These algorithms, by reducing the variance of the stochastic gradients, are able to guarantee convergence to the exact solution of the optimization problem with faster convergence than classical *SGD*. Over the past decade, many efficient variance-reduced methods have been proposed. Some popular examples of variance reduced algorithms are *SAG* (Schmidt et al., 2017), *SAGA* (Defazio et al., 2014), *SVRG* (Johnson & Zhang, 2013) and *SARAH* (Nguyen et al., 2017). For more examples of variance reduced methods, see Defazio (2016); Konečný et al. (2016); Gower et al. (2020); Khaled et al. (2020); Horváth et al. (2020); Cutkosky & Orabona (2020); Dubois-Taine et al. (2022); Sadiev et al. (2022).

Among the variance reduced methods, *SARAH* is of our interest in this work. Like the popular *SVRG*, *SARAH* algorithm is composed of two nested loops. In each outer loop $k \geq 1$, the gradient estimate $v_0 = \nabla P(w_{k-1})$ is set to be the full gradient. Subsequently, in the inner loop, at $t \geq 1$, a biased estimator $v_t$ is used and defined recursively as

$$v_t = \nabla f_i(w_t) - \nabla f_i(w_{t-1}) + v_{t-1}, \tag{2}$$

where $i \in [n]$ is a random sample selected at $t$.

A common characteristic of the popular variance reduced methods is that the step-size $\alpha$ in their update rule $w_{t+1} = w_t - \alpha v_t$ is constant (or diminishing with predetermined rules) and that depends on the characteristics of (1). An exception to this rule is variance reduced method with Barzilai-Borwein step-size, named *BB-SVRG* and *BB-SARAH* proposed in Tan et al. (2016) and Li & Giannakis (2019) respectively. These methods allow to use Barzilai-Borwein (*BB*) step-size rule to update the step-size once in every epoch; for more examples, see Li et al. (2020); Yang et al. (2021). There are also methods proposing approach of using local Lipschitz smoothness to derive an adaptive step-size (Liu et al., 2019b) with additional tunable parameters or leveraging *BB* step-size with averaging schemes to automatically determine the inner loop size (Li et al., 2020). However, these methods do not fully take advantage of the local geometry, and **a truly adaptive algorithm: adjusting step-size at every (inner) iteration and eliminating need of tuning any hyper-parameters, is yet to be developed in the stochastic variance reduced framework.** This is exactly the main contribution of this work, as we mentioned in previous section.

## 2 Motivation

With our primary focus on the design of a stochastic recursive algorithm with adaptive step-size, we discuss our motivation in this section.

A standard approach of tuning the step-size involves the painstaking grid search on a wide range of candidates. While more sophisticated methods can design a tuning plan, they often struggle for efficiency and/or require a considerable amount of computing resources.

More importantly, tuning step-size requires knowledge that is not readily available at a starting point $w_0 \in \mathcal{R}^d$, and choices of step-size could be heavily influenced by the curvature provided $\nabla^2 P(w_0)$. *What if a step-size has to be small due to a "sharp" curvature initially, which becomes "flat" afterwards?*

To see this is indeed the case for many machine learning problems, let us consider logistic regression for a binary classification problem, i.e., $f_i(w) = \log(1 + \exp(-y_i x_i^T w)) + \frac{\lambda}{2}\|w\|^2$, where $x_i \in \mathcal{R}^d$ is a feature vector, $y_i \in \{-1, +1\}$ is a ground truth, and the ERM problem is in the form of (1). It is easy to derive the local curvature of $P(w)$, defined by its Hessian in the form

$$\nabla^2 P(w) = \frac{1}{n}\sum_{i=1}^n \underbrace{\frac{\exp(-y_i x_i^T w)}{[1 + \exp(-y_i x_i^T w)]^2}}_{s_i(w)} x_i x_i^T + \lambda I. \tag{3}$$

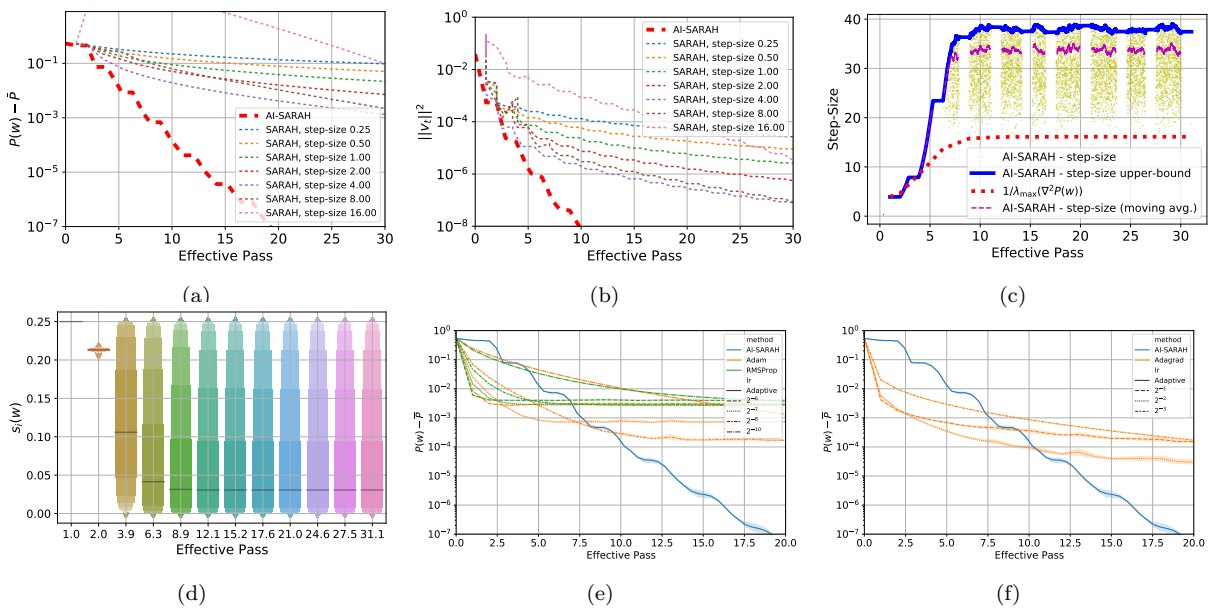

Figure 1: *AI-SARAH* vs. *SARAH*: **(a)** evolution of the optimality gap $P(w) - \bar{P}$ and **(b)** the squared norm of stochastic recursive gradient $\|v_t\|^2$; *AI-SARAH*: **(c)** evolution of the step-size, upper-bound, local Lipschitz smoothness and **(d)** distribution of $s_i$ of stochastic functions; **(e)** and **(f)** show the comparison of *AI-SARAH* with a constant step-size selection for non-variance reduced *ADAM* Kingma & Ba (2015), *RMSProp* Hinton et al. (2012); Bengio (2015), and *Adagrad* Duchi et al. (2011). Note that a larger step-size achieves quicker progress at the beginning but then stagnates. On the other hand, a smaller step-size slows down convergence, but the algorithms can achieve the solution of a better quality.
*Note: in (a), $\bar{P}$ is a lower bound of $P(w^*)$; in (c), the white spaces suggest full gradient computations at outer iterations; in (d), bars represent medians of $s_i$'s.*

Given that $\frac{a}{(1+a)^2} \le 0.25$ for any $a \ge 0$, one can immediately obtain the global bound on Hessian, i.e. $\forall w \in \mathcal{R}^d$ we have $\nabla^2 P(w) \preceq \frac{1}{4}\frac{1}{n}\sum_{i=1}^n x_i x_i^T + \lambda I$. Consequently, the parameter of global Lipschitz smoothness is $L = \frac{1}{4}\lambda_{\max}(\frac{1}{n}\sum_{i=1}^n x_i x_i^T) + \lambda$. It is well known that, with a constant step-size less than (or equal to) $\frac{1}{L}$, a convergence is guaranteed by many algorithms.

However, suppose the algorithm starts at a random $w_0$ (or at $\mathbf{0} \in \mathcal{R}^d$), this bound can be very tight. With more progress being made on approaching an optimal solution (or reducing the training error), it is likely that, for many training samples, $-y_i x_i^T w_t \ll 0$. An immediate implication is that $s_i(w_t)$ defined in (3) becomes smaller and hence the local curvature will be smaller as well. It suggests that, although a large initial step-size could lead to divergence, with more progress made by the algorithm, the parameter of local Lipschitz smoothness tends to be smaller and a larger step-size can be used. That being said, such a dynamic step-size cannot be well defined in the beginning, and a **fully adaptive approach needs to be developed.**

For illustration, we present the inspiring results of an experiment on *real-sim* dataset Chang & Lin (2011) with $\ell^2$-regularized logistic regression. Figures 1(a) and 1(b) compare the performance of classical *SARAH* with *AI-SARAH* in terms of the evolution of the optimality gap and the squared norm of recursive gradient. As is clear from the figure, *AI-SARAH* displays a significantly faster convergence per effective pass[1].

Now, let us discuss why this could happen. The distribution of $s_i$ as shown in Figured 1(d) indicates that: initially, all $s_i$'s are concentrated at 0.25; the median continues to reduce within a few effective passes on the training samples; eventually, it stabilizes somewhere below 0.05. Correspondingly, as presented in Figure 1(c), *AI-SARAH* starts with a conservative step-size dominated by the global Lipschitz smoothness, i.e., $1/\lambda_{max}(\nabla^2 P(w_0))$ (red dots); however, within 5 effective passes, the moving average (magenta dash) and

---

[1]The effective pass is defined as a complete pass on the training dataset. Each data sample is selected once per effective pass on average.

upper-bound (blue line) of the step-size start surpassing the red dots, and eventually stablize above the conservative step-size.

For classical *SARAH*, we configure the algorithm with different values of the fixed step-size, i.e., $\{2^{-2}, 2^{-1}, ..., 2^4\}$, and notice that $2^5$ leads to a divergence. On the other hand, *AI-SARAH* starts with a small step-size, yet achieves a faster convergence per effective pass with an eventual (moving average) step-size larger than $2^5$. Figures 1(e) and 1(f) show the comparison of *AI-SARAH* with a fixed step-size *ADAM* Kingma & Ba (2015), *RMSProp* Hinton et al. (2012); Bengio (2015), and *Adagrad* Duchi et al. (2011). The non-variance-reduced algorithms mentioned above can achieve faster initial convergence with a larger step-size. However, they achieve worse solutions in the given time budget. On the other hand, choosing a smaller step-size can lead to a better solution, but algorithms converge slower. In practice, one tunes the step-size and the step-size decay to achieve an acceptable performance.

## 3 Theoretical Analysis

In this section, we present the theoretical investigation on leveraging local Lipschitz smoothness to dynamically determine the step-size. We are trying to answer the main question: can we show convergence of using such an adaptive step-size and what are the benefits.

We present the theoretical framework in Algorithm 1 and refer to it as *Theoretical-AI-SARAH*. For the theoretical algorithm, we analyze two options for sampling functions:
**Option I.** - sampling $f_i$ uniformly at random, and
**Option II.** - importance sampling, where function $f_i$ is sampled with probability proportional to local $L_i$.

For brevity, we show the main results in the section and defer the full technical details to Appendix A.

---

**Algorithm 1** *Theoretical-AI-SARAH*

---

1: **Parameter:** Inner loop size $m$
2: **Initialize:** $\tilde{w}_0$
3: **for** $k = 1, 2, ...$ **do**
4:     $w_0 = \tilde{w}_{k-1}$
5:     $v_0 = \nabla P(w_0)$
6:     **for** $i \in [n]$ **do**
7:         Compute $L_i^0$ in the neighborhood of $w_0$ by (4)
8:     **end for**
9:     Compute $L^0 = \begin{cases} \max_{i \in [n]} L_i^0, & \text{Option I} \\ \frac{1}{n} \sum_{i=1}^n L_i^0, & \text{Option II} \end{cases}$ and set $\eta_0 = 1/L^0$
10:     **for** $t = 1, ..., m$ **do**
11:         $w_t = w_{t-1} - \eta_{t-1} v_{t-1}$
12:         Sample $i_t$ randomly from $[n]$ with probability $p_i^{t-1} = \begin{cases} \frac{1}{n}, & \text{Option I} \\ L_i^{t-1} / \sum_{j=1}^n L_j^{t-1}, & \text{Option II} \end{cases}$
13:         $v_t = v_{t-1} + \begin{cases} \nabla f_{i_t}(w_t) - \nabla f_{i_t}(w_{t-1}), & \text{Option I} \\ \frac{1}{np_i^{t-1}} \left( \nabla f_{i_t}(w_t) - \nabla f_{i_t}(w_{t-1}) \right), & \text{Option II} \end{cases}$
14:         **for** $i \in [n]$ **do**
15:             Compute $L_i^t$ in the neighborhood of $w_t$ by (4)
16:         **end for**
17:         Compute $L^t = \begin{cases} \max_{i \in [n]} L_i^t, & \text{Option I} \\ \frac{1}{n} \sum_{i=1}^n L_i^t, & \text{Option II} \end{cases}$
18:         $\eta_t = \min \left\{ \frac{1}{L^t}, \frac{L^{t-1}}{L^t} \eta_{t-1} \right\}$
19:     **end for**
20:     Set $\tilde{w}_k = w_t$ where $t$ is chosen with probability $q_t = \eta_t / \sum_{j=0}^m \eta_j$ from $\{0, 1, ..., m\}$
21: **end for**

---

For $t \geq 0, i_t \in [n]$, We assume $f_{i_t}$ is $L_i^t$-smooth on the line segment $\Delta_i^t = \{w \in \mathcal{R}^d \mid w = w_t - \eta v_t, \ \eta \in [0, \frac{1}{L_i^t}]\}$. Then, for Lines 7 and 15, we have

$$L_i^t = \max \|\nabla^2 f_i(w_t - \eta v_t)\|_2, \quad \text{where } \eta \in [0, \tfrac{1}{L_i^t}]. \tag{4}$$

The problem (4) essentially computes the largest eigenvalue of Hessian matrices on the defined line segment. Note that, (4) computes $L_i^t$ implicitly as it appears on both sides of the equation.

Having presented Algorithm 1, we can now show our main technical result in the following theorem.

**Theorem 3.1.** *Suppose $P$ is $\mu$-strongly convex, and each $f_i$ is convex and $L_i^t$-smooth on the line segment $\Delta_i^t = \{w \in \mathcal{R}^d \mid w = w_t - \eta v_t, \ \eta \in [0, \frac{1}{L_i^t}]\}$. For $k \geq 1$, let us define*

$$\sigma_m^k = \left( \frac{1}{\mu \mathcal{H}} + \frac{\eta_0 L^0}{2 - \eta_0 L^0} \right), \tag{5}$$

*where $\mathcal{H} = \sum_{t=0}^m \eta_t$, and select $m$ and $\eta$ such that $\sigma_m^l < 1$, Algorithm 1 converges as follows*

$$\mathbb{E}[\|\nabla P(\tilde{w}_k)\|^2] \leq \left( \prod_{l=1}^k \sigma_m^l \right) \|\nabla P(\tilde{w}_0)\|^2.$$

*Remark* 3.2. The convergence result of classical SARAH Nguyen et al. (2017) algorithm for strongly convex case has a similar form of $\sigma$ as (5) but with $\mathcal{H}_{SARAH} = \alpha(m+1)$, where $\alpha \leq \frac{1}{L}$ is a constant step-size. In this case, $L$ is the global smoothness parameter of each $f_i, i \in [n]$. Now, as (4) defines $L_i^t$ to be the parameter of smoothness only on a line segment, we trivially have that

$$L_i^t \overset{(4)}{=} \max_{\eta \in [0, \frac{1}{L_i^t}]} \|\nabla^2 f_i(w_t - \eta v_t)\| \leq \max_{w \in \mathcal{R}^d} \|\nabla^2 f_i(w)\| \leq L.$$

Thus, $\mathcal{H} = \sum_{t=0}^m \eta_t = \sum_{t=0}^m \frac{1}{L^t} \geq \sum_{t=0}^m \frac{1}{L} = \frac{m+1}{L} \geq \alpha(m+1) = \mathcal{H}_{SARAH}$. Then, it is clear that, Algorithm 1 can achieve a faster convergence than classical *SARAH*.

By Theorem 3.1 and Remark 3.2, we show that, in theory, by leveraging local Lipschitz smoothness, Algorithm 1 is guaranteed to converge and can even achieve a faster convergence than classical *SARAH* if local geometry permits.

With that being said, we note that Algorithm 1 requires the computations of the largest eigenvalues of Hessian matrices on the line segment for each $f_i$ at every outer and inner iterations. In general, such computations would be too expensive, and thus would keep one from solving Problem (1) efficiently in practice.

In the next section, we will present our main contribution of the paper, the practical algorithm, *AI-SARAH*. **It does not only eliminate the expensive computations in Algorithm 1, but also eliminate efforts of tuning hyper-parameters.**

## 4 AI-SARAH

We present the practical algorithm, *AI-SARAH*, in Algorithm 2. At every iteration, instead of incurring expensive costs on computing the parameters of local Lipshitz smoothness for all $f_i$ in Algorithm 1, Algorithm 2 estimates the local smoothness by approximately solving the sub-problem for only one $f_i$, i.e., $\min_{\alpha>0} \xi_t(\alpha)$, with a minimal extra cost in addition to computing stochastic gradient, i.e., $\nabla f_i$. Also, by approximately solving the sub-problem, Algorithm 2 implicitly computes the step-size, i.e., $\alpha_{t-1}$ at $t \geq 1$. Please note that, on Line 9, $b > 0$ is the mini-batch size. Let us remark that AI-SARAH samples function $f_i$ uniformly, and here for simplicity, we do not focus on proposing a practical version with an importance sampling.

In Algorithm 2, we adopts an adaptive upper-bound with exponential smoothing. To be specific, the upper-bound is updated with exponential smoothing on harmonic mean of the approximate solutions to the sub-problems, which also keeps track of the estimates of local Lipschitz smoothness.

In the following sections, we will present the details on the design of *AI-SARAH*.

We note that **this algorithm is fully adaptive and requires no efforts of tuning, and can be implemented easily**. Notice that $\beta$ is treated as a smoothing factor in updating the upper-bound of the step-size, and the default setting is $\beta = 0.999$. There exists one hyper-parameter in Algorithm 2, $\gamma$, which defines the early stopping criterion on Line 8, and the default setting is $\gamma = \frac{1}{32}$. We will show later in this section that, the performance of this algorithm is not sensitive to the choices of $\gamma$, and this is true regardless of the problems (i.e., regularized/non-regularized logistic regression and different datasets.)

---

**Algorithm 2** *AI-SARAH*

---

1: **Parameter:** $0 < \gamma < 1$ (default $\frac{1}{32}$), $\beta = 0.999$
2: **Initialize:** $\tilde{w}_0$
3: **Set:** $\alpha_{max} = \infty$
4: **for** k = 1, 2, ... **do**
5:     $w_0 = \tilde{w}_{k-1}$
6:     $v_0 = \nabla P(w_0)$
7:     $t = 1$
8:     **while** $\|v_t\|^2 \geq \gamma \|v_0\|^2$ **do**
9:         Select random mini-batch $S_t$ from $[n]$ uniformly with $|S_t| = b$
10:         $\tilde{\alpha}_{t-1} \approx \arg\min_{\alpha>0} \xi_t(\alpha)$
11:         **if** $k = 1$ and $t = 1$ **then**
12:             $\delta_t^k = \frac{1}{\tilde{\alpha}_{t-1}}$
13:         **else**
14:             $\delta_t^k = \beta \delta_{t-1}^k + (1 - \beta) \frac{1}{\tilde{\alpha}_{t-1}}$
15:         **end if**
16:         $\alpha_{max} = \frac{1}{\delta_t^k}$
17:         $\alpha_{t-1} = \min\{\tilde{\alpha}_{t-1}, \alpha_{max}\}$
18:         $w_t = w_{t-1} - \alpha_{t-1} v_{t-1}$
19:         $v_t = \nabla f_{S_t}(w_t) - \nabla f_{S_t}(w_{t-1}) + v_{t-1}$
20:         $t = t + 1$
21:     **end while**
22:     Set $\delta_0^{k+1} = \delta_t^k$
23:     Set $\tilde{w}_k = w_t$
24: **end for**

---

### 4.1 Estimate Local Lipschitz Smoothness

In the previous section, we showed that Algorithm 1 computes the parameters of local Lipschitz smoothness, and it can be very expensive and thus prohibited in practice. To avoid the expensive cost, one can estimate the local Lipschitz smoothness instead of computing an exact parameter. Then, the question is how to estimate the parameter of local Lipschitz smoothness in practice.

**Can we use line-search?** The standard approach to estimate local Lipschitz smoothness is to use backtracking line-search. Recall *SARAH*'s update rule, i.e., $w_t = w_{t-1} - \alpha_{t-1} v_{t-1}$, where $v_{t-1}$ is a stochastic recursive gradient. The standard procedure is to apply line-search on function $f_{i_t}(w_{t-1} - \alpha v_{t-1})$. However, the main issue is that $-v_{t-1}$ is not necessarily a descent direction.

**AI-SARAH sub-problem.** Define the sub-problem[2] (as shown on Line 10 of Algorithm 2) as

$$\min_{\alpha>0} \xi_t(\alpha) = \min_{\alpha>0} \|\nabla f_{i_t}(w_{t-1} - \alpha v_{t-1}) - \nabla f_{i_t}(w_{t-1}) + v_{t-1}\|^2, \tag{6}$$

---

[2]For the sake of simplicity, we use $f_{i_t}$ instead of $f_{S_t}$.

where $t \geq 1$ denotes an inner iteration and $i_t$ indexes a random sample selected at $t$. We argue that, by (approximately) solving (6), we can have a good estimate of the parameters of the local Lipschitz smoothness.

To illustrate this setting, we denote $L_i^t$ the parameter of local Lipschitz smoothness prescribed by $f_{i_t}$ at $w_{t-1}$. Let us focus on a simple quadratic function $f_{i_t}(w) = \frac{1}{2}(x_{i_t}^T w - y_{i_t})^2$. Let $\tilde{\alpha}$ be the optimal step-size along direction $-v_{t-1}$, i.e. $\tilde{\alpha} = \arg\min_\alpha f_{i_t}(w_{t-1} - \alpha v_{t-1})$. Then, the closed form solution of $\tilde{\alpha}$ can be easily derived as $\tilde{\alpha} = \frac{x_{i_t}^T w_{t-1} - y_{i_t}}{x_{i_t}^T v_{t-1}}$, whose value can be positive, negative, bounded or unbounded.

On the other hand, one can compute the step-size implicitly by solving (6) and obtain $\alpha_{t-1}^i$, i.e., $\alpha_{t-1}^i = \arg\min_\alpha \xi_t(\alpha)$. Then, we have $\alpha_{t-1}^i = \frac{1}{x_{i_t}^T x_{i_t}}$, which is exactly $\frac{1}{L_i^t}$.

**To put it simply, as quadratic function has a constant Hessian, solving (6) gives exactly $\frac{1}{L_i^t}$. For general (strongly) convex functions, if $\nabla^2 f_{i_t}(w_{t-1})$, does not change too much locally, we can still have a good estimate of $1/L_i^t$ in direction of $v_{t-1}$ by solving (6) approximately**. To see that, let us assume that we can approximate the difference of gradients as follows

$$\nabla f_{i_t}(w_{t-1} - \alpha v_{t-1}) - \nabla f_{i_t}(w_{t-1}) \approx -\alpha \nabla^2 f_{i_t}(w_{t-1}) v_{t-1}. \tag{7}$$

Then

$$\arg\min_\alpha \| -\alpha \nabla^2 f_{i_t}(w_{t-1}) v_{t-1} + v_{t-1} \|^2 = \frac{v_{t-1}^T \nabla^2 f_{i_t}(w_{t-1}) v_{t-1}}{v_{t-1}^T \nabla^2 f_{i_t}(w_{t-1}) \nabla^2 f_{i_t}(w_{t-1}) v_{t-1}}. \tag{8}$$

The reciprocal of the last expression could be seen as the curvature of $f_{i_t}(w_{t-1})$ in direction $v_{t-1}$.

Based on a good estimate of $L_i^t$, we can then obtain the estimate of the local Lipschitz smoothness of $P(w_{t-1})$. And, that is $L^t = \frac{1}{n} \sum_{i=1}^n L_i^t = \frac{1}{n} \sum_{i=1}^n \frac{1}{\alpha_{t-1}^i}$. Clearly, if a step-size in the algorithm is selected as $1/L^t$, then a harmonic mean of the sequence of the step-size's, computed for various component functions could serve as a good adaptive upper-bound on the step-size computed in the algorithm. More details of intuition for the adaptive upper-bound can be found in Appendix B.2.

## 4.2 Compute Step-size and Upper-bound

On Line 10 of Algorithm 2, the sub-problem is a one-dimensional minimization problem, which can be approximately solved by Newton method. Specifically in Algorithm 2, we compute *one-step Newton* at $\alpha = 0$, and that is

$$\tilde{\alpha}_{t-1} = -\frac{\xi_t'(0)}{|\xi_t''(0)|}. \tag{9}$$

Note that, for convex function in general, (9) gives an approximate solution; for functions in particular forms such as quadratic ones, (9) gives an exact solution.

In order to give an insight why one-step Newton at $\alpha = 0$ could make sense, let us first evaluate the key quantities in (9). We have

$$\xi_t'(0) = -v_{t-1}^T \nabla^2 f_{i_t}(w_{t-1}) v_{t-1},$$
$$\xi_t''(0) = \nabla^3 f_{i_t}(w_{t-1})[v_{t-1}, v_{t-1}, v_{t-1}] + v_{t-1}^T \nabla^2 f_{i_t}(w_{t-1}) \nabla^2 f_{i_t}(w_{t-1}) v_{t-1},$$

where $\nabla^3 f_{i_t}(w_{t-1})[v_{t-1}, v_{t-1}, v_{t-1}]$ is the 3rd derivative of $f_{i_t}$ multiplied by $v_{t-1}$. If we assume that the Hessian is not changing much, then this quantity can be ignored and (9) becomes equal to (8).

The procedure prescribed in (9) can be implemented very efficiently, and **it does not require any extra (stochastic) gradient computations if compared with classical $SARAH$.** The only extra cost per iteration is to perform two backward passes, i.e., one pass for $\xi_t'(0)$ and the other for $\xi_t''(0)$; see Appendix B.2 for implementation details.

As shown on Lines 11-16, 22 of Algorithm 2, $\alpha_{max}$ is updated at every inner iteration. Specifically, **the algorithm starts without an upper bound (i.e., $\alpha_{max} = \infty$ on Line 3); as $\tilde{\alpha}_{t-1}$ being computed at every**

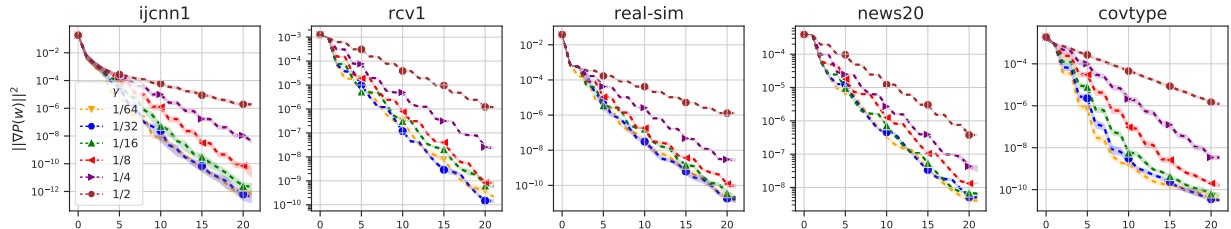

Figure 2: Evolution of $\|\nabla P(w)\|^2$ for $\gamma \in \{\frac{1}{64}, \frac{1}{32}, \frac{1}{16}, \frac{1}{8}, \frac{1}{4}, \frac{1}{2}\}$ on the **regularized** case.

Table 1: Summary of Datasets from Chang & Lin (2011).

| Dataset | # features | $n$ (# Train) | # Test | % Sparsity |
|---|---|---|---|---|
| *ijcnn1*[1] | 22 | 49,990 | 91,701 | 40.91 |
| *rcv1*[1] | 47,236 | 20,242 | 677,399 | 99.85 |
| *real-sim*[2] | 20,958 | 54,231 | 18,078 | 99.76 |
| *news20*[2] | 1,355,191 | 14,997 | 4,999 | 99.97 |
| *covtype*[2] | 54 | 435,759 | 145,253 | 77.88 |

[1] dataset has default training/testing samples.
[2] dataset is randomly split by 75%-training & 25%-testing.

$t \geq 1$, we employs the exponential smoothing on the harmonic mean of $\{\tilde{\alpha}_{t-1}\}$ to update the upper-bound. For $k \geq 1$ and $t \geq 1$, we define $\alpha_{max} = \frac{1}{\delta_t^k}$, where

$$\delta_t^k = \begin{cases} \frac{1}{\tilde{\alpha}_{t-1}}, & k = 1, t = 1 \\ \beta\delta_{t-1}^k + (1 - \beta)\frac{1}{\tilde{\alpha}_{t-1}}, & otherwise \end{cases}$$

and $0 < \beta < 1$. We default $\beta = 0.999$ in Algorithm 2. At the end of the $k$th outer loop, denoted $t = T$, we let $\delta_0^{k+1} = \delta_T^k$; see Appendix B.2 for details on the design of the adaptive upper-bound.

### 4.3 Choice of $\gamma$

We perform a sensitivity analysis on different choices of $\gamma$. Figures 2 shows the evolution of the squared norm of full gradient, i.e., $\|\nabla P(w)\|^2$, for $\ell^2$-regularized logistic regression on binary classification problems; see non-regularized case and extended results in Appendix B. It is clear that the performance of $\gamma$'s, where, $\gamma \in \{1/8, 1/16, 1/32, 1/64\}$, is consistent with only marginal improvement by using a smaller value. We default $\gamma = 1/32$ in Algorithm 2.

## 5 Numerical Experiment

In this section, we present the empirical study on the performance of *AI-SARAH* (see Algorithm 2). For brevity, we present a subset of experiments in the main paper, and defer the full experimental results and implementation details[3] in Appendix B.

The problems we consider in the experiment are $\ell^2$-regularized logistic regression for binary classification problems; see Appendix B for non-regularized case. Given a training sample $(x_i, y_i)$ indexed by $i \in [n]$, the component function $f_i$ is in the form $f_i(w) = \log(1 + \exp(-y_i x_i^T w)) + \frac{\lambda}{2}\|w\|^2$, where $\lambda = \frac{1}{n}$ for the $\ell^2$-regularized case and $\lambda = 0$ for the non-regularized case. The datasets chosen for the experiments are *ijcnn1, rcv1, real-sim, news20* and *covtype*. Table 1 shows the basic statistics of the datasets. More details and additional datasets can be found in Appendix B.

---

[3]See code at `https://github.com/shizheng-rlfresh/ai_sarah`.

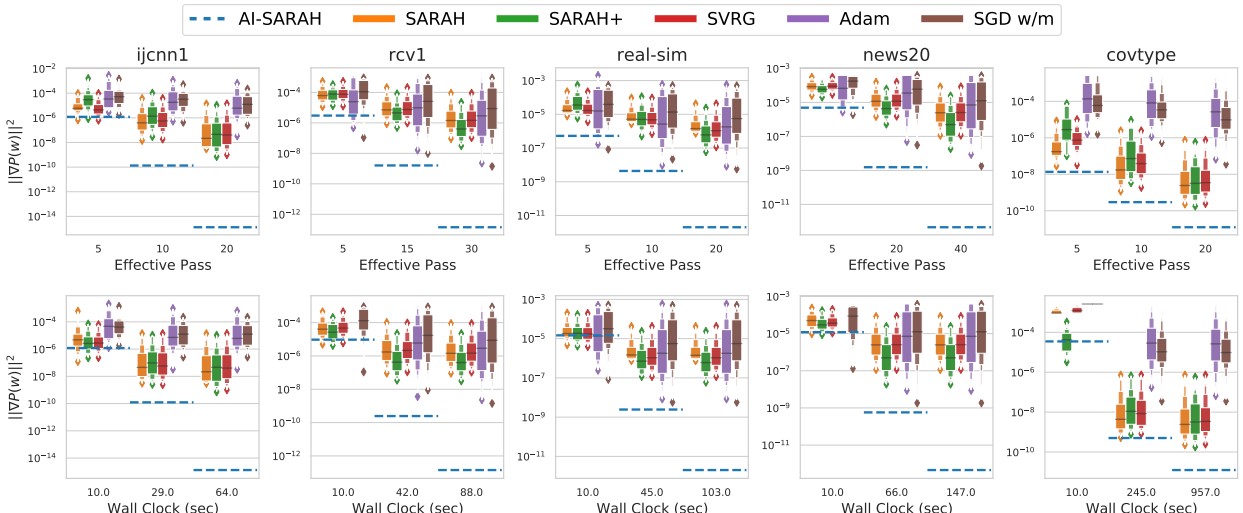

Figure 3: Running minimum per effective pass (top row) and wall clock time (bottom row) of $\|\nabla P(w)\|^2$ between other algorithms with all hyper-parameters configurations and *AI-SARAH* for the **regularized** case.

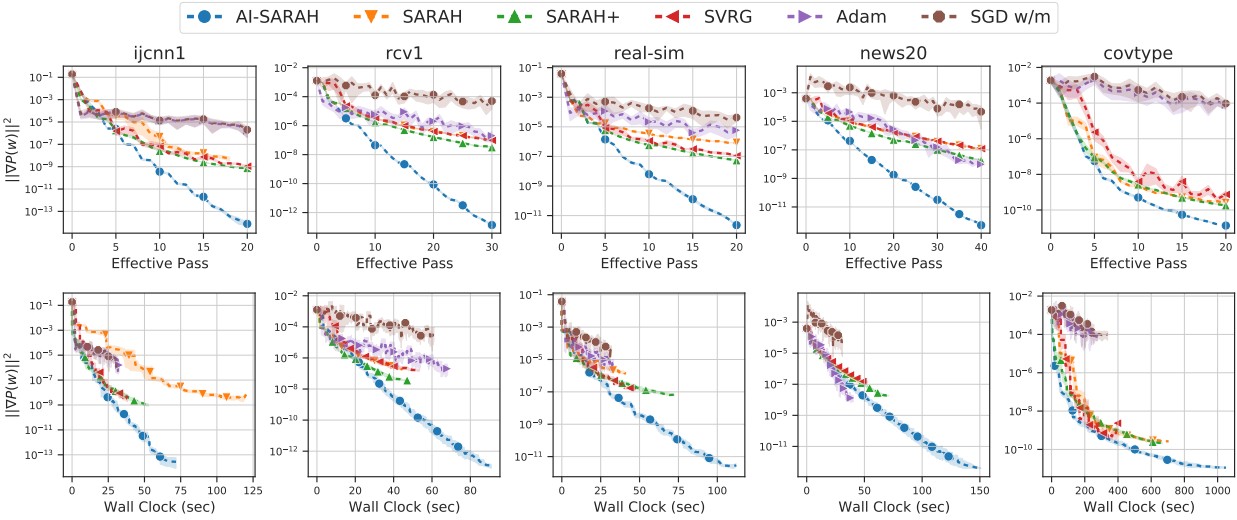

Figure 4: Evolution of $\|\nabla P(w)\|^2$ for the **regularized** case by effective pass (top row) and wall clock time (bottom row).

We compare *AI-SARAH* with *SARAH*, *SARAH+*, *SVRG* (Johnson & Zhang, 2013), *ADAM* (Kingma & Ba, 2015) and *SGD* with Momentum (Sutskever et al., 2013; Loizou & Richtárik, 2020; 2017). **While *AI-SARAH* does not require hyper-parameter tuning, we fine-tune each of the other algorithms, which yields $\approx 5,000$ runs in total for each dataset and case.**

To be specific, we perform an extensive search on hyper-parameters: (1) *ADAM* and *SGD* with Momentum (*SGD* w/m) are tuned with different values of the (initial) step-size and schedules to reduce the step-size; (2) *SARAH* and *SVRG* are tuned with different values of the (constant) step-size and inner loop size; (3) *SARAH+* is tuned with different values of the (constant) step-size and early stopping parameter. (See Appendix B for detailed tuning plan and the selected hyper-parameters.)

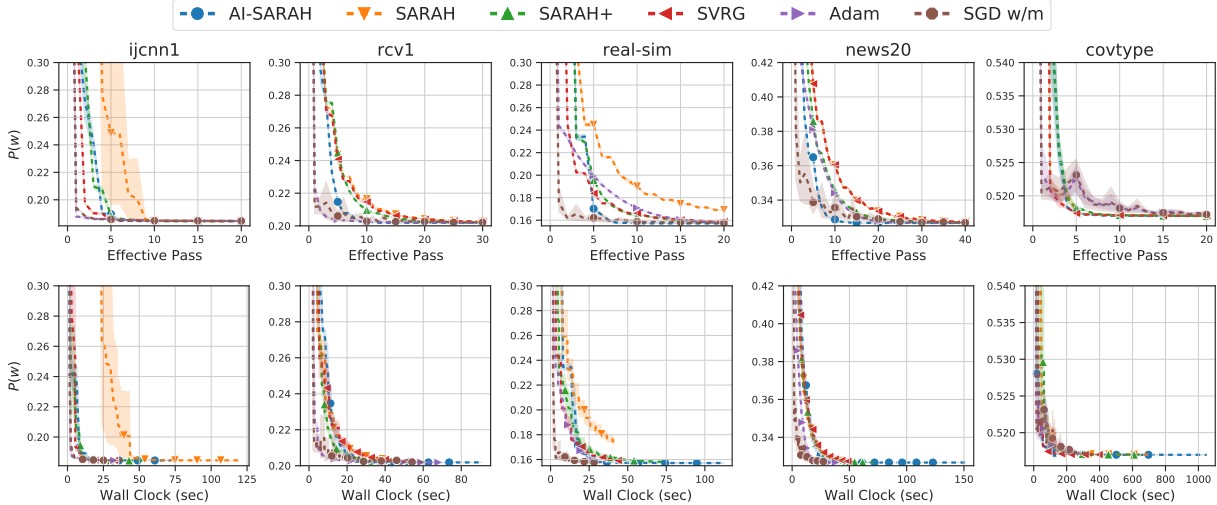

Figure 5: Evolution of $P(w)$ for the **regularized** case by effective pass (top row) and wall clock time (bottom row).

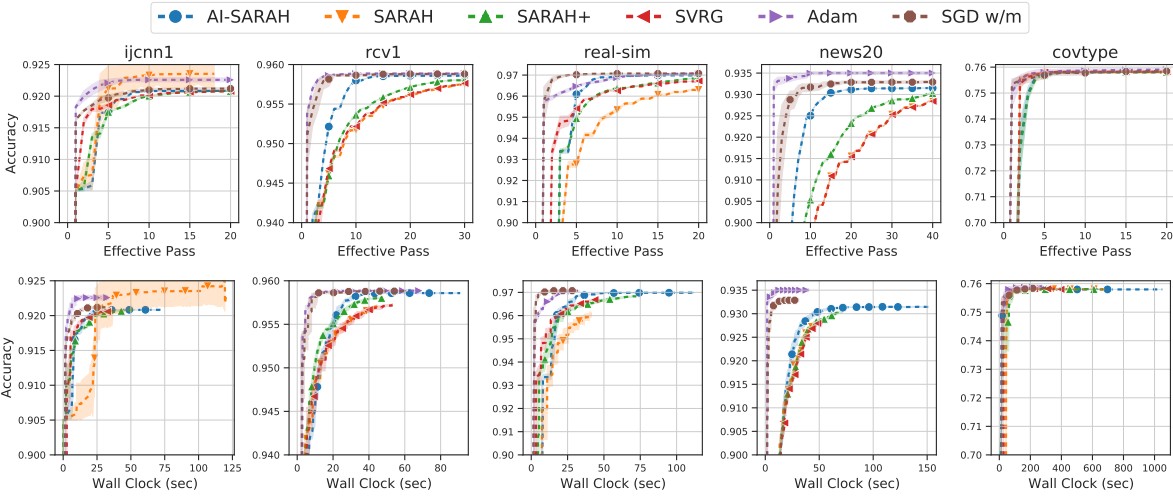

Figure 6: Running maximum of testing accuracy for the **regularized** case by effective pass (top row) and wall clock time (bottom row).

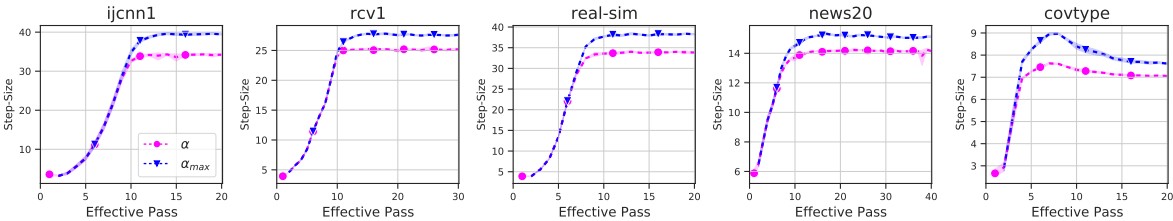

Figure 7: Evolution of *AI-SARAH*'s step-size $\alpha$ and upper-bound $\alpha_{max}$ for the **regularized** case.

Figure 3 shows the minimum $\|\nabla P(w)\|^2$ achieved at a few points of effective passes and wall clock time horizon. It is clear that, *AI-SARAH*'s practical speed of convergence is faster than the other algorithms in most cases. Here, we argue that, if given an optimal implementation of *AI-SARAH* (just as that of *ADAM* and other built-in optimizer in Pytorch[4] ), it is likely that our algorithm can be accelerated.

By selecting the fine-tuned hyper-parameters of all other algorithms, we compare them with *AI-SARAH* and show the results in Figures 4-6. For these experiments, we use 10 distinct random seeds to initialize $w$ and generate stochastic mini-batches. And, we use the marked dashes to represent the average and filled areas for 97% confidence intervals.

Figure 4 presents the evolution of $\|\nabla P(w)\|^2$. Obviously from the figure, *AI-SARAH* exhibits the strongest performance in terms of converging to a stationary point: by effective pass, the consistently large gaps are displayed between *AI-SARAH* and the rest; by wall clock time, we notice that *AI-SARAH* achieves the smallest $\|\nabla P(w)\|^2$ at the same time point. This validates our design, that is to leverage local Lipschitz smoothness and achieve a faster convergence than *SARAH* and *SARAH+*.

In terms of minimizing the finite-sum functions, Figure 5 shows that, by effective pass, *AI-SARAH* consistently outperforms *SARAH* and *SARAH+* on all of the datasets with a possible exception on *covtype* dataset. By wall clock time, *AI-SARAH* yields a competitive performance on all of the datasets, and it delivers a stronger performance on *ijcnn1* and *real-sim* than *SARAH*.

For completeness of illustration on the performance, we show the testing accuracy in Figure 6. Clearly, fine-tuned *ADAM* dominates the competition. However, *AI-SARAH* outperforms the other variance reduced methods on most of the datasets from both effective pass and wall clock time perspectives, and achieves the similar levels of accuracy as *ADAM* does on *rcv1*, *real-sim* and *covtype* datasets.

Having illustrated the strong performance of *AI-SARAH*, we continue the presentation by showing the trajectories of the adaptive step-size and upper-bound in Figure 7. This figure clearly shows that why *AI-SARAH* can achieve such a strong performance, especially on the convergence to a stationary point. As mentioned in the previous sections, the adaptivity is driven by the local Lipschitz smoothness. As shown in Figure 7, *AI-SARAH* starts with conservative step-size and upper-bound, both of which continue to increase while the algorithm progresses towards a stationary point. After a few effective passes, we observe: the step-size and upper-bound are stablized due to $\lambda > 0$ (and hence strong convexity). In Appendix B, we can see that, as a result of the function being unregularized, and thus likely non-strongly convex, the step-size and upper-bound could be continuously increasing.

## 6 Conclusion

In this paper, we propose *AI-SARAH*, a practical variant of stochastic recursive gradient methods. The idea of design is simple yet powerful: by taking advantage of local Lipschitz smoothness, the step-size can be dynamically determined. With intuitive illustration and implementation details, we show how *AI-SARAH* can efficiently estimate local Lipschitz smoothness and how it can be easily implemented in practice. Our algorithm is tune-free and adaptive at full scale. With extensive numerical experiment, we demonstrate that, without (tuning) any hyper-parameters, it delivers a competitive performance compared with *SARAH(+)*, *ADAM* and other first-order methods, all equipped with fine-tuned hyper-parameters.

### Acknowledgments

The work of P.R. was partially supported by the KAUST Baseline Research Fund Scheme and by the SDAIA-KAUST Center of Excellence in Data Science and Artificial Intelligence. The work of M.T. was partially supported by the MBZUAI-WIS research grant.

---

[4]Please see `https://pytorch.org/docs/stable/optim.html` for Pytorch built-in optimizers.

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

# Appendix

The Appendix is organized as follows. In Section A, we present the technical details of theoretical analysis in Section 3 of the main paper. In Section B, we present extended details on the design, implementation and results of our numerical experiments.

## A    Technical Results and Proofs

We consider finite-sum optimization problem

$$\min_{w \in \mathcal{R}^d} \left[ P(w) \stackrel{\text{def}}{=} \frac{1}{n} \sum_{i=1}^{n} f_i(w) \right]. \tag{10}$$

**Assumption A.1.** For $t \geq 0$, each $f_i$ is $L_i^t$-smooth on the line segment $\Delta = \left\{ w \in \mathcal{R}^d \mid w = w_t - \eta v_t, \forall \eta \in \left[ 0, \frac{1}{L_i^t} \right] \right\}$ and convex. For simplicity, we denote

$$L^t = \max_{i \in [n]} L_i^t, \ \bar{L}^t = \frac{1}{n} \sum_{i}^{n} L_i^t, \ \bar{L}_M = \max_{t \in \{0,1,\dots,m\}} \bar{L}^t.$$

Note that in Section 3 of the main paper, we use $L^t$ universally for both maximum and average value of parameters of local Lipschitz smoothness. In this section, as we will present Algorithm 1 in two specific forms: importance sampling version (see Algorithm 3) and uniform sampling version (see Algorithm 4), we use a different notation on the average, i.e., $\bar{L}^t = \frac{1}{n} \sum_{i}^{n} L_i^t$.

**Assumption A.2.** Function $P$ is $\mu$-strongly convex.

**Definition A.3.** Fix a outer loop $k \geq 1$ and consider Algorithms 1, 3 and 4 with an inner loop size $m$, we define a discrete probability distribution at $t \geq 1$ for all $i \in [n]$, $p_i^t = \frac{L_i^t}{\sum_{i=1}^{n} L_i^t}$, and probabilities $q_t$ for all $t \geq 0$, $q_t = \frac{\eta_t}{\mathcal{H}}$, where $\mathcal{H} = \sum_{j=0}^{m} \eta_j$.

### A.1    Theoretical-AI-SARAH with Importance Sampling

We present the importance sampling algorithm in Algorithm 3. Now, let us start by presenting the following lemmas that are extending the lemmas in Nguyen et al. (2017) for the SARAH algorithm.

**Lemma A.4.** *Consider $v_t$ defined in Algorithm 3. Then for any $t \geq 1$ in Algorithm 3, it holds that*

$$\mathbb{E}[\|\nabla P(w_t) - v_t\|^2] = \sum_{j=1}^{t} \mathbb{E}[\|v_j - v_{j-1}\|^2] - \sum_{j=1}^{t} \mathbb{E}[\|\nabla P(w_j) - \nabla P(w_{j-1})\|^2].$$

*Proof.* Let $\mathbb{E}_j$ denote the expectation by conditioning on the information $w_0, w_1, \dots, w_j$ as well as $v_0, v_1, \dots, v_{j-1}$. Then,

$$\begin{aligned}
\mathbb{E}_j[\|\nabla P(w_j) - v_j\|^2] &= \mathbb{E}_j \left[ \| \left( \nabla P(w_{j-1}) - v_{j-1} \right) + \left( \nabla P(w_j) - \nabla P(w_{j-1}) \right) - \left( v_j - v_{j-1} \right) \|^2 \right] \\
&= \mathbb{E}_j[\|\nabla P(w_{j-1}) - v_{j-1}\|^2] + \|\nabla P(w_j) - \nabla P(w_{j-1})\|^2 \\
&\quad + \mathbb{E}_j[\|v_j - v_{j-1}\|^2] \\
&\quad + 2 \langle \nabla P(w_{j-1}) - v_{j-1}, \nabla P(w_j) - \nabla P(w_{j-1}) \rangle \\
&\quad - 2 \langle \nabla P(w_{j-1}) - v_{j-1}, \mathbb{E}_j[v_j - v_{j-1}] \rangle \\
&\quad - 2 \langle \nabla P(w_j) - \nabla P(w_{j-1}), \mathbb{E}_j[v_j - v_{j-1}] \rangle \\
&= \mathbb{E}_j[\|\nabla P(w_{j-1}) - v_{j-1}\|^2] - \|\nabla P(w_j) - \nabla P(w_{j-1})\|^2 \\
&\quad + \mathbb{E}_j[\|v_j - v_{j-1}\|^2], \tag{11}
\end{aligned}$$

---

**Algorithm 3** *Theoretical-AI-SARAH with Importance Sampling*

---

1: **Parameter:** Inner loop size $m$
2: **Initialize:** $\tilde{w}_0$
3: **for** $k = 1, 2, ...$ **do**
4:      $w_0 = \tilde{w}_{k-1}$
5:      $v_0 = \nabla P(w_0)$
6:      **for** $i \in [n]$ **do**
7:          $L_i^0 = \max_{\eta \in [0, \frac{1}{L_i^0}]} \|\nabla^2 f_i(w_0 - \eta v_0)\|$
8:      **end for**
9:      $\bar{L}^0 = \frac{1}{n} \sum_{i=1}^n L_i^0$ and $\eta_0 = \frac{1}{\bar{L}^0}$
10:     **for** $t = 1, ..., m$ **do**
11:        $w_t = w_{t-1} - \eta_{t-1} v_{t-1}$
12:        Sample $i_t$ from $[n]$ with probability $p_i^{t-1}$
13:        $v_t = v_{t-1} + \frac{1}{np_i^{t-1}} \left(\nabla f_{i_t}(w_t) - \nabla f_{i_t}(w_{t-1})\right)$
14:        **for** $i \in [n]$ **do**
15:            $L_i^t = \max_{\eta \in [0, \frac{1}{L_i^t}]} \|\nabla^2 f_i(w_t - \eta v_t)\|$
16:        **end for**
17:        $\bar{L}^t = \frac{1}{n} \sum_{i=1}^n L_i^t$
18:        $\eta_t = \min \left\{ \frac{1}{\bar{L}^t}, \frac{\bar{L}^{t-1}}{\bar{L}^t} \eta_{t-1} \right\}$
19:     **end for**
20:     **Set** $\tilde{w}_k = w_t$ where $t$ is chosen with probability $q_t$ from $\{0, 1, ..., m\}$
21: **end for**

---

where the last equality follows from

$$
\begin{aligned}
\mathbb{E}_j[v_j - v_{j-1}] &= \mathbb{E}_j\big[\frac{1}{np_{i_j}^{j-1}} \left(\nabla f_{i_j}(w_j) - \nabla f_{i_j}(w_{j-1})\right)\big] \\
&= \sum_{i_j}^n \frac{p_{i_j}^{j-1}}{np_{i_j}^{j-1}} \left(\nabla f_{i_j}(w_j) - \nabla f_{i_j}(w_{j-1})\right) \\
&= \nabla P(w_j) - \nabla P(w_{j-1}).
\end{aligned}
\tag{12}
$$

By taking expectation of (11), we have

$$
\begin{aligned}
\mathbb{E}[\|\nabla P(w_j) - v_j\|^2] = \mathbb{E}[\|\nabla P(w_{j-1}) - v_{j-1}\|^2] &- \mathbb{E}[\|\nabla P(w_j) - \nabla P(w_{j-1})\|^2] \\
&+ \mathbb{E}[\|v_j - v_{j-1}\|^2].
\end{aligned}
$$

By summing it over $j = 1, ..., t$ and note that $\|\nabla P(v_0) - v_0\|^2 = 0$, we have

$$
\mathbb{E}[\|\nabla P(w_t) - v_t\|^2] = \sum_{j=1}^t \mathbb{E}[\|v_j - v_{j-1}\|^2] - \sum_{j=1}^t \mathbb{E}[\|\nabla P(w_j) - \nabla P(w_{j-1})\|^2].
$$

$\square$

**Lemma A.5.** *Fix a outer loop $k \geq 1$ and consider Algorithm 3 with $\eta_t \leq 1/\bar{L}^t$ for any $t \in [m]$. Under Assumption A.1,*

$$
\sum_{t=0}^m \frac{\eta_t}{2} \mathbb{E}[\|\nabla P(w_t)\|^2] \leq \mathbb{E}[P(w_0) - P(w^*)] + \sum_{t=0}^m \frac{\eta_t}{2} \mathbb{E}[\|\nabla P(w_t) - v_t\|^2].
$$

*Proof.* By Assumption A.1 and the update rule $w_t = w_{t-1} - \eta_{t-1} v_{t-1}$ of Algorithm 3, we obtain

$$P(w_t) \leq P(w_{t-1}) - \eta_{t-1} \langle \nabla P(w_{t-1}), v_{t-1} \rangle + \frac{\bar{L}^{t-1}}{2} \eta_{t-1}^2 \|v_{t-1}\|^2$$

$$= P(w_{t-1}) - \frac{\eta_{t-1}}{2} \|\nabla P(w_{t-1})\|^2 + \frac{\eta_{t-1}}{2} \|\nabla P(w_{t-1}) - v_{t-1}\|^2$$

$$- \left( \frac{\eta_{t-1}}{2} - \frac{\bar{L}^{t-1}}{2} \eta_{t-1}^2 \right) \|v_{t-1}\|^2,$$

where, in the equality above, we use the fact that $\langle a, b \rangle = \frac{1}{2}(\|a\|^2 + \|b\|^2 - \|a - b\|^2)$.

By assuming that $\eta_{t-1} \leq \frac{1}{\bar{L}^{t-1}}$, it holds that $\left(1 - \bar{L}^{t-1} \eta_{t-1}\right) \geq 0$, $\forall t \in [m]$. Thus,

$$\frac{\eta_{t-1}}{2} \|\nabla P(w_{t-1})\|^2 \leq [P(w_{t-1}) - P(w_t)] + \frac{\eta_{t-1}}{2} \|\nabla P(w_{t-1}) - v_{t-1}\|^2$$

$$- \frac{\eta_{t-1}}{2} \left(1 - \bar{L}^{t-1} \eta_{t-1}\right) \|v_{t-1}\|^2.$$

By taking expectations

$$\mathbb{E}[\frac{\eta_{t-1}}{2} \|\nabla P(w_{t-1})\|^2] \leq \mathbb{E}[P(w_{t-1})] - \mathbb{E}[P(w_t)] + \frac{\eta_{t-1}}{2} \mathbb{E}[\|\nabla P(w_{t-1}) - v_{t-1}\|^2]$$

$$- \frac{\eta_{t-1}}{2} \left(1 - \bar{L}^{t-1} \eta_t\right) \mathbb{E}[\|v_{t-1}\|^2]$$

$$\overset{\eta_{t-1} \leq \frac{1}{\bar{L}^{t-1}}}{\leq} \mathbb{E}[P(w_{t-1})] - \mathbb{E}[P(w_t)] + \frac{\eta_{t-1}}{2} \mathbb{E}[\|\nabla P(w_{t-1}) - v_{t-1}\|^2].$$

Summing over $t = 1, 2, \ldots, m+1$, we have

$$\sum_{t=1}^{m+1} \frac{\eta_{t-1}}{2} \mathbb{E}[\|\nabla P(w_{t-1})\|^2] \leq \sum_{t=1}^{m+1} \mathbb{E}[P(w_{t-1}) - P(w_t)] + \sum_{t=1}^{m+1} \frac{\eta_{t-1}}{2} \mathbb{E}[\|\nabla P(w_{t-1}) - v_{t-1}\|^2]$$

$$= \mathbb{E}[P(w_0) - P(w_{m+1})] + \sum_{t=1}^{m+1} \frac{\eta_{t-1}}{2} \mathbb{E}[\|\nabla P(w_{t-1}) - v_{t-1}\|^2]$$

$$\leq \mathbb{E}[P(w_0) - P(w_*)] + \sum_{t=1}^{m+1} \frac{\eta_{t-1}}{2} \mathbb{E}[\|\nabla P(w_{t-1}) - v_{t-1}\|^2],$$

where the last inequality holds since $w^*$ is the global minimizer of $P$.

The last expression can be equivalently written as

$$\sum_{t=0}^{m} \frac{\eta_t}{2} \mathbb{E}[\|\nabla P(w_t)\|^2] \leq \mathbb{E}[P(w_0) - P(w_*)] + \sum_{t=0}^{m} \frac{\eta_t}{2} \mathbb{E}[\|\nabla P(w_t) - v_t\|^2],$$

which completes the proof. $\qquad \square$

**Lemma A.6.** *Consider Algorithm 3 with $\eta_t = \min\left\{ \frac{1}{L^t}, \frac{\bar{L}^{t-1}}{\bar{L}^t} \eta_{t-1} \right\}$. Suppose $f_i$ is convex for all $i \in [n]$. Then, under Assumption A.1, for any $t \geq 1$,*

$$\mathbb{E}[\|\nabla P(w_t) - v_t\|^2] \leq \left( \frac{\eta_0 \bar{L}^0}{2 - \eta_0 \bar{L}^0} \right) \mathbb{E}[\|v_0\|^2].$$

*Proof.*

$$\mathbb{E}_j\left[\|v_j\|^2\right] \leq \mathbb{E}_j\left[\|v_{j-1} - \frac{1}{np_i^{j-1}}\left(\nabla f_{i_j}(w_{j-1}) - \nabla f_{i_j}(w_j)\right)\|^2\right]$$

$$= \|v_{j-1}\|^2 + \mathbb{E}_j\left[\frac{1}{\left(np_i^{j-1}\right)^2}\|\nabla f_{i_j}(w_{j-1}) - \nabla f_{i_j}(w_j)\|^2\right]$$

$$- \mathbb{E}_j\left[\frac{2}{\eta_{j-1}np_i^{j-1}}\left\langle\nabla f_{i_j}(w_{j-1}) - \nabla f_{i_j}(w_j), w_{j-1} - w_j\right\rangle\right]$$

$$\leq \|v_{j-1}\|^2 + \mathbb{E}_j\left[\frac{1}{(np_i^{j-1})^2}\|\nabla f_{i_j}(w_{j-1}) - \nabla f_{i_j}(w_j)\|^2\right]$$

$$- \mathbb{E}_j\left[\frac{2}{\eta_{j-1}np_i^{j-1}L_{i_j}^{j-1}}\|\nabla f_{i_j}(w_{j-1}) - \nabla f_{i_j}(w_j)\|^2\right].$$

For each outer loop $k \geq 1$, it holds that $L_{i_j}^{j-1} = np_i^{j-1}\bar{L}^{j-1}$. Thus,

$$\mathbb{E}_j[\|v_j\|^2] \leq \|v_{j-1}\|^2 + \mathbb{E}_j\left[\left\|\frac{1}{np_i^{j-1}}\left(\nabla f_{i_j}(w_{j-1}) - \nabla f_{i_j}(w_j)\right)\right\|^2\right]$$

$$- \frac{2}{\eta_{j-1}\bar{L}^{j-1}}\mathbb{E}_j\left[\left\|\frac{1}{np_i^{j-1}}\left(\nabla f_{i_j}(w_{j-1}) - \nabla f_{i_j}(w_j)\right)\right\|^2\right]$$

$$= \|v_{j-1}\|^2 + \left(1 - \frac{2}{\eta_{j-1}\bar{L}^{j-1}}\right)\mathbb{E}_j\left[\left\|\frac{1}{np_i^{j-1}}\left(\nabla f_{i_j}(w_{j-1}) - \nabla f_{i_j}(w_j)\right)\right\|^2\right]$$

$$= \|v_{j-1}\|^2 + \left(1 - \frac{2}{\eta_{j-1}\bar{L}^{j-1}}\right)\mathbb{E}_j\left[\|v_j - v_{j-1}\|^2\right]$$

$$\leq \|v_{j-1}\|^2 + \left(1 - \frac{2}{\eta_{j-1}\bar{L}^{j-1}}\right)\mathbb{E}_j\left[\|v_j - v_{j-1}\|^2\right].$$

By rearranging, taking expectations again, and assuming that $\eta_{j-1} < 2/\bar{L}^{j-1}$ for any $j$ from 1 to $t+1$

$$\mathbb{E}[\|v_j - v_{j-1}\|^2] \leq \mathbb{E}\left[\left(\frac{\eta_{j-1}\bar{L}^{j-1}}{2 - \eta_{j-1}\bar{L}^{j-1}}\right)\left[\|v_{j-1}\|^2 - \|v_j\|^2\right]\right].$$

By summing the above inequality over $j = 1, \ldots, t$ $(t \geq 1)$, we have

$$\sum_{j=1}^{t}\mathbb{E}[\|v_j - v_{j-1}\|^2] \leq \sum_{j=1}^{t}\mathbb{E}\left[\left(\frac{\eta_{j-1}\bar{L}^{j-1}}{2 - \eta_{j-1}\bar{L}^{j-1}}\right)\left[\|v_{j-1}\|^2 - \|v_j\|^2\right]\right]$$

$$= \left(\frac{\eta_0\bar{L}^0}{2 - \eta_0\bar{L}^0}\right)\mathbb{E}\left[\|v_0\|^2\right] - \left(\frac{\eta_{t-1}\bar{L}^{t-1}}{2 - \eta_{t-1}\bar{L}^{t-1}}\right)\mathbb{E}\left[\|v_t\|^2\right]$$

$$- \sum_{j=1}^{t-1}\mathbb{E}\left[\left(\frac{\eta_{j-1}\bar{L}^{j-1}}{2 - \eta_{j-1}\bar{L}^{j-1}} - \frac{\eta_j\bar{L}^j}{2 - \eta_j\bar{L}^j}\right)\|v_j\|^2\right]$$

$$\leq \left(\frac{\eta_0\bar{L}^0}{2 - \eta_0\bar{L}^0}\right)\mathbb{E}\left[\|v_0\|^2\right]. \tag{13}$$

Now, by using Lemma A.4 and (13), we obtain

$$\mathbb{E}[\|\nabla P(w_t) - v_t\|^2] \le \sum_{j=1}^{t} \mathbb{E}\left[\|v_j - v_{j-1}\|^2\right] \le \left(\frac{\eta_0 \bar{L}^0}{2 - \eta_0 \bar{L}^0}\right) \mathbb{E}\left[\|v_0\|^2\right].$$

$\square$

Using the above lemmas, we can present one of our main results in the following theorem.

**Theorem A.7.** *Suppose that Assumptions A.1, A.2, holds. Let us define*

$$\bar{\sigma}_m^k = \left(\frac{1}{\mu\mathcal{H}} + \frac{\eta_0 \bar{L}^0}{2 - \eta_0 \bar{L}^0}\right),$$

*and select $m$ and $\eta$ such that $\bar{\sigma}_m^k < 1$. Then, Algorithm 3 converges as follows*

$$\mathbb{E}[\|\nabla P(\tilde{w}_k)\|^2] \le \left(\prod_{l=1}^{k} \bar{\sigma}_m^l\right) \|\nabla P(\tilde{w}_0)\|^2.$$

*Proof.* Since $v_0 = \nabla P(w_0)$ implies $\|\nabla P(w_0) - v_0\|^2 = 0$, then by Lemma A.6, we obtain

$$\sum_{t=0}^{m} \frac{\eta_t}{\mathcal{H}} \mathbb{E}[\|\nabla P(w_t) - v_t\|^2] \le \left(\frac{\eta_0 \bar{L}^0}{2 - \eta_0 \bar{L}^0}\right) \mathbb{E}[\|v_0\|^2].$$

Combine this with Lemma A.5, we have that

$$\sum_{t=0}^{m} \frac{\eta_t}{\mathcal{H}} \mathbb{E}[\|\nabla P(w_t)\|^2] \le \frac{2}{\mathcal{H}} \mathbb{E}[P(w_0) - P(w_*)] + \sum_{t=0}^{m} \frac{\eta_t}{\mathcal{H}} \mathbb{E}[\|\nabla P(w_t) - v_t\|^2]$$

$$\le \frac{2}{\mathcal{H}} \mathbb{E}[P(w_0) - P(w_*)] + \frac{\eta_0 \bar{L}^0}{2 - \eta_0 \bar{L}^0} \mathbb{E}[\|v_0\|^2].$$

Since we consider one outer loop, with $k \ge 1$, we have $v_0 = \nabla P(w_0) = \nabla P(\tilde{w}_{k-1})$ and $\tilde{w}_k = w_t$, where $t$ is drawn at random from $\{0, 1, \dots, m\}$ with probabilities $q_t$. Therefore, the following holds,

$$\mathbb{E}[\|\nabla P(\tilde{w}_k)\|^2] = \sum_{t=0}^{m} \sum_{t=0}^{m} \frac{\eta_t}{\mathcal{H}} \mathbb{E}[\|\nabla P(w_t)\|^2]$$

$$\le \frac{2}{\mathcal{H}} \mathbb{E}[P(\tilde{w}_{k-1}) - P(w_*)] + \frac{\eta_0 \bar{L}^0}{2 - \eta_0 \bar{L}^0} \mathbb{E}[\|\nabla P(\tilde{w}_{k-1})\|^2]$$

$$\le \left(\frac{1}{\mu\mathcal{H}} + \frac{\eta_0 \bar{L}^0}{2 - \eta_0 \bar{L}^0}\right) \mathbb{E}[\|\nabla P(\tilde{w}_{k-1})\|^2].$$

Let us define $\bar{\sigma}_m^k = \left(\frac{1}{\mu\mathcal{H}} + \frac{\eta_0 \bar{L}^0}{2 - \eta_0 \bar{L}^0}\right)$, then the above expression can be written as

$$\mathbb{E}[\|\nabla P(\tilde{w}_k)\|^2] \le \bar{\sigma}_m^k \mathbb{E}[\|\nabla P(\tilde{w}_{k-1})\|^2].$$

By expanding the recurrence, we obtain

$$\mathbb{E}[\|\nabla P(\tilde{w}_k)\|^2] \le \left(\prod_{l=1}^{k} \bar{\sigma}_m^l\right) \|\nabla P(\tilde{w}_0)\|^2.$$

This completes the proof. $\square$

## A.2 Theoretical-AI-SARAH with Uniform Sampling

We present the uniform sampling algorithm in Algorithm 4. Now, let us start by presenting the following lemmas.

---

**Algorithm 4** *Theoretical-AI-SARAH with Uniform Sampling*

---

1: **Parameter:** Inner loop size $m$
2: **Initialize:** $\tilde{w}_0$
3: **for** k = 1, 2, ... **do**
4:  $w_0 = \tilde{w}_{k-1}$
5:  $v_0 = \nabla P(w_0)$
6:  **for** $i \in [n]$ **do**
7:    $L_i^0 = \max_{\eta \in [0, \frac{1}{L_i^0}]} \|\nabla^2 f_i(w_0 - \eta v_0)\|$
8:  **end for**
9:  $L^0 = \max_{i \in [n]} L_i^0$ and $\eta_0 = \frac{1}{L^0}$
10:  **for** t = 1, ... , m **do**
11:    $w_t = w_{t-1} - \eta_{t-1} v_{t-1}$
12:    Sample $i_t$ uniformly at random from $[n]$
13:    $v_t = v_{t-1} + \nabla f_{i_t}(w_t) - \nabla f_{i_t}(w_{t-1})$
14:    **for** $i \in [n]$ **do**
15:      $L_i^t = \max_{\eta \in [0, \frac{1}{L_i^t}]} \|\nabla^2 f_i(w_t - \eta v_t)\|$
16:    **end for**
17:    $L^t = \max_{i \in [n]} L_i^t$
18:    $\eta_t = \min \left\{ \frac{1}{L^t}, \frac{L^{t-1}}{L^t} \eta_{t-1} \right\}$
19:  **end for**
20:  **Set** $\tilde{w}_k = w_t$ where $t$ is chosen with probability $q_t$ from $\{0, 1, ..., m\}$
21: **end for**

---

**Lemma A.8.** *Consider $v_t$ defined in Algorithm 4. Then for any $t \geq 1$ in Algorithm 4, it holds that*

$$\mathbb{E}[\|\nabla P(w_t) - v_t\|^2] = \sum_{j=1}^{t} \mathbb{E}[\|v_j - v_{j-1}\|^2] - \sum_{j=1}^{t} \mathbb{E}[\|\nabla P(w_j) - \nabla P(w_{j-1})\|^2].$$

*Proof.* The proof is the same as that of Lemma A.4 except that we have $p_{i_j}^{j-1} = \frac{1}{n}$ in (12) for uniform sampling. □

**Lemma A.9.** *Fix a outer loop $k \geq 1$ and consider Algorithm 4 with $\eta_t \leq 1/L^t$ for any $t \in [m]$. Under Assumption A.1,*

$$\sum_{t=0}^{m} \frac{\eta_t}{2} \mathbb{E}[\|\nabla P(w_t)\|^2] \leq \mathbb{E}[P(w_0) - P(w^*)] + \sum_{t=0}^{m} \frac{\eta_t}{2} \mathbb{E}[\|\nabla P(w_t) - v_t\|^2].$$

*Proof.* By Assumption A.1 and the update rule $w_t = w_{t-1} - \eta_{t-1} v_{t-1}$ of Algorithm 4, we obtain

$$P(w_t) \leq P(w_{t-1}) - \eta_{t-1} \langle \nabla P(w_{t-1}), v_{t-1} \rangle + \frac{L^{t-1}}{2} \eta_{t-1}^2 \|v_{t-1}\|^2$$

$$= P(w_{t-1}) - \frac{\eta_{t-1}}{2} \|\nabla P(w_{t-1})\|^2 + \frac{\eta_{t-1}}{2} \|\nabla P(w_{t-1}) - v_{t-1}\|^2$$

$$- \left( \frac{\eta_{t-1}}{2} - \frac{L^{t-1}}{2} \eta_{t-1}^2 \right) \|v_{t-1}\|^2,$$

where, in the equality above, we use the fact that $\langle a, b \rangle = \frac{1}{2}(\|a\|^2 + \|b\|^2 - \|a - b\|^2)$.

By assuming that $\eta_{t-1} \leq \frac{1}{L^{t-1}}$, it holds $\left(1 - L^{t-1}\eta_{t-1}\right) \geq 0$, $\forall t \in [m]$. Thus,

$$\frac{\eta_{t-1}}{2}\|\nabla P(w_{t-1})\|^2 \leq [P(w_{t-1}) - P(w_t)] + \frac{\eta_{t-1}}{2}\|\nabla P(w_{t-1}) - v_{t-1}\|^2$$
$$- \frac{\eta_{t-1}}{2}\left(1 - L^{t-1}\eta_{t-1}\right)\|v_{t-1}\|^2.$$

By taking expectations

$$\mathbb{E}[\frac{\eta_{t-1}}{2}\|\nabla P(w_{t-1})\|^2] \leq \mathbb{E}[P(w_{t-1})] - \mathbb{E}[P(w_t)] + \frac{\eta_{t-1}}{2}\mathbb{E}[\|\nabla P(w_{t-1}) - v_{t-1}\|^2]$$
$$- \frac{\eta_{t-1}}{2}\left(1 - L^{t-1}\eta_{t-1}\right)\mathbb{E}[\|v_{t-1}\|^2]$$
$$\overset{\eta_{t-1} \leq \frac{1}{L^{t-1}}}{\leq} \mathbb{E}[P(w_{t-1})] - \mathbb{E}[P(w_t)] + \frac{\eta_{t-1}}{2}\mathbb{E}[\|\nabla P(w_{t-1}) - v_{t-1}\|^2].$$

Summing over $t = 1, 2, \ldots, m+1$, we have

$$\sum_{t=1}^{m+1} \frac{\eta_{t-1}}{2}\mathbb{E}[\|\nabla P(w_{t-1})\|^2] \leq \sum_{t=1}^{m+1} \mathbb{E}[P(w_{t-1}) - P(w_t)] + \sum_{t=1}^{m+1} \frac{\eta_{t-1}}{2}\mathbb{E}[\|\nabla P(w_{t-1}) - v_{t-1}\|^2]$$
$$= \mathbb{E}[P(w_0) - P(w_{m+1})] + \sum_{t=1}^{m+1} \frac{\eta_{t-1}}{2}\mathbb{E}[\|\nabla P(w_{t-1}) - v_{t-1}\|^2]$$
$$\leq \mathbb{E}[P(w_0) - P(w_*)] + \sum_{t=1}^{m+1} \frac{\eta_{t-1}}{2}\mathbb{E}[\|\nabla P(w_{t-1}) - v_{t-1}\|^2],$$

where the last inequality holds since $w^*$ is the global minimum of $P$.

The last expression can be equivalently written as

$$\sum_{t=0}^{m} \frac{\eta_t}{2}\mathbb{E}[\|\nabla P(w_t)\|^2] \leq \mathbb{E}[P(w_0) - P(w_*)] + \sum_{t=0}^{m} \frac{\eta_t}{2}\mathbb{E}[\|\nabla P(w_t) - v_t\|^2],$$

which completes the proof. $\qquad \square$

**Lemma A.10.** *Consider Algorithm 4 with $\eta_t = \min\left\{\frac{1}{L^t}, \frac{L^{t-1}}{L^t}\eta_{t-1}\right\}$. Suppose $f_i$ is convex for all $i \in [n]$. Then, under Assumption A.1, for any $t \geq 1$,*

$$\mathbb{E}[\|\nabla P(w_t) - v_t\|^2] \leq \left(\frac{\eta_0 L^0}{2 - \eta_0 L^0}\right)\mathbb{E}[\|v_0\|^2].$$

*Proof.*

$$\mathbb{E}_{i_j}\left[\|v_j\|^2\right] \leq \mathbb{E}_{i_j}\left[\|v_{j-1} - \left(\nabla f_{i_j}(w_{j-1}) - \nabla f_{i_j}(w_j)\right)\|^2\right]$$
$$= \|v_{j-1}\|^2 + \mathbb{E}_{i_j}\left[\|\nabla f_{i_j}(w_{j-1}) - \nabla f_{i_j}(w_j)\|^2\right]$$
$$- \mathbb{E}_{i_j}\left[\frac{2}{\eta_{j-1}}\left\langle \nabla f_{i_j}(w_{j-1}) - \nabla f_{i_j}(w_j), w_{j-1} - w_j\right\rangle\right]$$
$$\leq \|v_{j-1}\|^2 + \mathbb{E}_{i_j}\left[\|\nabla f_{i_j}(w_{j-1}) - \nabla f_{i_j}(w_j)\|^2\right]$$
$$- \mathbb{E}_{i_j}\left[\frac{2}{\eta_{j-1}L_{i_j}^{j-1}}\|\nabla f_{i_j}(w_{j-1}) - \nabla f_{i_j}(w_j)\|^2\right].$$

For each outer loop $k \geq 1$, it holds that $L_{i_j}^{j-1} \leq L^{j-1}$. Thus,

$$
\begin{aligned}
\mathbb{E}_{i_j}[\|v_j\|^2] &\leq \|v_{j-1}\|^2 + \mathbb{E}_{i_j}\left[\left\|\nabla f_{i_j}(w_{j-1}) - \nabla f_{i_j}(w_j)\right\|^2\right] \\
&\quad - \frac{2}{\eta_{j-1}L^{j-1}}\mathbb{E}_{i_j}\left[\left\|\nabla f_{i_j}(w_{j-1}) - \nabla f_{i_j}(w_j)\right\|^2\right] \\
&= \|v_{j-1}\|^2 + \left(1 - \frac{2}{\eta_{j-1}L^{j-1}}\right)\mathbb{E}_{i_j}\left[\left\|\nabla f_{i_j}(w_{j-1}) - \nabla f_{i_j}(w_j)\right\|^2\right] \\
&= \|v_{j-1}\|^2 + \left(1 - \frac{2}{\eta_{j-1}L^{j-1}}\right)\mathbb{E}_{i_j}\left[\|v_j - v_{j-1}\|^2\right]. \\
&\leq \|v_{j-1}\|^2 + \left(1 - \frac{2}{\eta_{j-1}L^{j-1}}\right)\mathbb{E}_{i_j}\left[\|v_j - v_{j-1}\|^2\right].
\end{aligned}
$$

By rearranging, taking expectations again, and assuming that $\eta_{j-1} < 2/L^{j-1}$ for any $j$ from 1 to $t+1$,

$$
\mathbb{E}[\|v_j - v_{j-1}\|^2] \leq \mathbb{E}\left[\left(\frac{\eta_{j-1}L^{j-1}}{2 - \eta_{j-1}L^{j-1}}\right)\left[\|v_{j-1}\|^2 - \|v_j\|^2\right]\right].
$$

By summing the above inequality over $j = 1, \ldots, t$ $(t \geq 1)$, we have

$$
\begin{aligned}
\sum_{j=1}^{t}\mathbb{E}[\|v_j - v_{j-1}\|^2] &\leq \sum_{j=1}^{t}\mathbb{E}\left[\left(\frac{\eta_{j-1}L^{j-1}}{2 - \eta_{j-1}L^{j-1}}\right)\left[\|v_{j-1}\|^2 - \|v_j\|^2\right]\right] \\
&= \left(\frac{\eta_0 L^0}{2 - \eta_0 L^0}\right)\mathbb{E}\left[\|v_0\|^2\right] - \left(\frac{\eta_{t-1}L^{t-1}}{2 - \eta_{t-1}L^{t-1}}\right)\mathbb{E}\left[\|v_t\|^2\right] \\
&\quad - \sum_{j=1}^{t-1}\mathbb{E}\left[\left(\frac{\eta_{j-1}L^{j-1}}{2 - \eta_{j-1}L^{j-1}} - \frac{\eta_j L^j}{2 - \eta_j L^j}\right)\|v_j\|^2\right] \\
&\leq \left(\frac{\eta_0 L^0}{2 - \eta_0 L^0}\right)\mathbb{E}\left[\|v_0\|^2\right]. \quad\quad\quad (14)
\end{aligned}
$$

Now, by using Lemma A.8 and (14), we obtain

$$
\begin{aligned}
\mathbb{E}[\|\nabla P(w_t) - v_t\|^2] &\leq \sum_{j=1}^{t}\mathbb{E}\left[\|v_j - v_{j-1}\|^2\right] \\
&\leq \left(\frac{\eta_0 L^0}{2 - \eta_0 L^0}\right)\mathbb{E}\left[\|v_0\|^2\right].
\end{aligned}
$$

$\square$

Using the above lemmas, we can present one of our main results in the following theorem.

**Theorem A.11.** *Suppose that Assumption A.1, A.2, holds. Let us define*

$$
\sigma_m^k = \left(\frac{1}{\mu\mathcal{H}} + \frac{\eta_0 L^0}{2 - \eta_0 L^0}\right),
$$

*and select $m$ and $\eta$ such that $\sigma_m^k < 1$. Then, Algorithm 4 converges as follows*

$$
\mathbb{E}[\|\nabla P(\tilde{w}_k)\|^2] \leq \left(\prod_{l=1}^{k}\sigma_m^l\right)\|\nabla P(\tilde{w}_0)\|^2.
$$

*Proof.* Since $v_0 = \nabla P(w_0)$ implies $\|\nabla P(w_0) - v_0\|^2 = 0$, then by Lemma A.10, we obtain:

$$\sum_{t=0}^{m} \frac{\eta_t}{\mathcal{H}} \mathbb{E}[\|\nabla P(w_t) - v_t\|^2] \leq \left( \frac{\eta_0 L^0}{2 - \eta_0 L^0} \right) \mathbb{E}[\|v_0\|^2].$$

Combine this with Lemma A.9, we have

$$\sum_{t=0}^{m} \frac{\eta_t}{\mathcal{H}} \mathbb{E}[\|\nabla P(w_t)\|^2] \leq \frac{2}{\mathcal{H}} \mathbb{E}[P(w_0) - P(w_*)] + \sum_{t=0}^{m} \frac{\eta_t}{\mathcal{H}} \mathbb{E}[\|\nabla P(w_t) - v_t\|^2]$$

$$\leq \frac{2}{\mathcal{H}} \mathbb{E}[P(w_0) - P(w_*)] + \frac{\eta_0 L^0}{2 - \eta_0 L^0} \mathbb{E}[\|v_0\|^2].$$

Since we consider one outer loop, with $k \geq 1$, we have $v_0 = \nabla P(w_0) = \nabla P(\tilde{w}_{k-1})$ and $\tilde{w}_k = w_t$, where $t$ is drawn at random from $\{0, 1, \ldots, m\}$ with probabilities $q_t$. Therefore, the following holds,

$$\mathbb{E}[\|\nabla P(\tilde{w}_k)\|^2] = \sum_{t=0}^{m} \sum_{t=0}^{m} \frac{\eta_t}{\mathcal{H}} \mathbb{E}[\|\nabla P(w_t)\|^2]$$

$$\leq \frac{2}{\mathcal{H}} \mathbb{E}[P(\tilde{w}_{k-1}) - P(w_*)] + \frac{\eta_0 L^0}{2 - \eta_0 L^0} \mathbb{E}[\|\nabla P(\tilde{w}_{k-1})\|^2]$$

$$\leq \left( \frac{1}{\mu \mathcal{H}} + \frac{\eta_0 L^0}{2 - \eta_0 L^0} \right) \mathbb{E}[\|\nabla P(\tilde{w}_{k-1})\|^2].$$

Let us use $\sigma_m^k = \left( \frac{1}{\mu \mathcal{H}} + \frac{\eta_0 L^0}{2 - \eta_0 L^0} \right)$, then the above expression can be written as

$$\mathbb{E}[\|\nabla P(\tilde{w}_k)\|^2] \leq \sigma_m^k \mathbb{E}[\|\nabla P(\tilde{w}_{k-1})\|^2].$$

By expanding the recurrence, we obtain

$$\mathbb{E}[\|\nabla P(\tilde{w}_k)\|^2] \leq \left( \prod_{l=1}^{k} \sigma_m^l \right) \|\nabla P(\tilde{w}_0)\|^2.$$

This completes the proof. $\qquad \square$

# B    Extended details on Numerical Experiment

In this section, we present the extended details of the design, implementation and results of the numerical experiments.

## B.1    Problem and Data

The machine learning tasks studied in the experiment are binary classification problems. As a common practice in the empirical research of optimization algorithms, the *LIBSVM* datasets[5] are chosen to define the tasks. Specifically, **we selected** 10 **popular binary class datasets: *ijcnn1, rcv1, news20, covtype, real-sim, a1a, gisette, w1a, w8a* and *mushrooms*** (see Table 2 for basic statistics of the datasets). Please note that these datasets do not contain personally identifiable information or offensive content.

Table 2: Summary of Datasets from Chang & Lin (2011).

| Dataset | $d-1$ (# feature) | $n$ (# Train) | $n_{test}$ (# Test) | % Sparsity |
|---------|-------------------|---------------|---------------------|------------|
| *ijcnn1*[1] | 22 | 49,990 | 91,701 | 40.91 |
| *rcv1*[1] | 47,236 | 20,242 | 677,399 | 99.85 |
| *news20*[2] | 1,355,191 | 14,997 | 4,999 | 99.97 |
| *covtype*[2] | 54 | 435,759 | 145,253 | 77.88 |
| *real-sim*[2] | 20,958 | 54,231 | 18,078 | 99.76 |
| *a1a*[1] | 123 | 1,605 | 30,956 | 88.73 |
| *gisette*[1] | 5,000 | 6,000 | 1,000 | 0.85 |
| *w1a*[1] | 300 | 2,477 | 47,272 | 96.11 |
| *w8a*[1] | 300 | 49,749 | 14,951 | 96.12 |
| *mushrooms*[2] | 112 | 6,093 | 2,031 | 81.25 |

[1] dataset has default training/testing samples.
[2] dataset is randomly split by 75%-training & 25%-testing.

### B.1.1    Data Pre-Processing

Let $(\chi_i, y_i)$ be a training (or testing) sample indexed by $i \in [n]$ (or $i \in [n_{test}]$), where $\chi_i \in \mathcal{R}^{d-1}$ is a feature vector and $y_i$ is a label. We pre-processed the data such that $\chi_i$ is of a unit length in Euclidean norm and $y_i \in \{-1, +1\}$.

### B.1.2    Model and Loss Function

The selected model, $h_i : \mathcal{R}^d \mapsto \mathcal{R}$, is in the linear form

$$h_i(\omega, \varepsilon) = \chi_i^T \omega + \varepsilon, \quad \forall i \in [n], \tag{15}$$

where $\omega \in \mathcal{R}^{d-1}$ is a weight vector and $\varepsilon \in \mathcal{R}$ is a bias term.

For simplicity of notation, from now on, we let $x_i \stackrel{\text{def}}{=} [\chi_i^T \; 1]^T \in \mathcal{R}^d$ be an augmented feature vector, $w \stackrel{\text{def}}{=} [\omega^T \; \varepsilon]^T \in \mathcal{R}^d$ be a parameter vector, and $h_i(w) = x_i^T w$ for $i \in [n]$.

Given a training sample indexed by $i \in [n]$, the loss function is defined as a logistic regression

$$f_i(w) = \log(1 + \exp(-y_i h_i(w)) + \frac{\lambda}{2} \|w\|^2. \tag{16}$$

In (16), $\frac{\lambda}{2}\|w\|^2$ is the $\ell^2$-regularization of a particular choice of $\lambda > 0$, where we used $\lambda = \frac{1}{n}$ in the experiment; for the non-regularized case, $\lambda$ was set to 0. Accordingly, the finite-sum minimization problem we aimed to solve is defined as

$$\min_{w \in \mathcal{R}^d} \left\{ P(w) \stackrel{\text{def}}{=} \frac{1}{n} \sum_{i=1}^n f_i(w) \right\}. \tag{17}$$

---

[5]*LIBSVM* datasets are available at `https://www.csie.ntu.edu.tw/~cjlin/libsvmtools/datasets/`.

Note that (17) is a convex function. For the $\ell^2$-regularized case, i.e., $\lambda = 1/n$ in (16), (17) is $\mu$-strongly convex and $\mu = \frac{1}{n}$. However, without the $\ell^2$-regularization, i.e., $\lambda = 0$ in (16), (17) is $\mu$-strongly convex if and only if there there exists $\mu > 0$ such that $\nabla^2 P(w) \succeq \mu I$ for $w \in \mathcal{R}^d$ (provided $\nabla P(w) \in \mathcal{C}$).

### B.2 Algorithms

This section provides the implementation details[6] of the algorithms, practical consideration, and discussions.

#### B.2.1 Tune-free *AI-SARAH*

In Section 4 of the main paper, we introduced $AI\text{-}SARAH$ (see Algorithm 2), a tune-free and fully adaptive algorithm. **The implementation of Algorithm 2 was quite straightforward, and we highlight the implementation of Line** 10 **with details**: for logistic regression, the one-dimensional (constrained optimization) sub-problem $\min_{\alpha>0} \xi_t(\alpha)$ can be approximately solved by computing the Newton step at $\alpha = 0$, i.e., $\tilde{\alpha}_{t-1} = -\frac{\xi_t'(0)}{|\xi_t''(0)|}$. This can be easily implemented with automatic differentiation in Pytorch[7], and only two additional backward passes w.r.t $\alpha$ is needed. For function in some particular form, such as a linear least square loss function, an exact solution in closed form can be easily derived.

As mentioned in Section 4, we have an adaptive upper-bound, i.e., $\alpha_{max}$, in the algorithm. To be specific, the algorithm starts without an upper-bound, i.e., $\alpha_{max} = \infty$ on Line 3 of Algorithm 2. Then, $\alpha_{max}$ is updated per (inner) iteration. Recall in Section 4, $\alpha_{max}$ is computed as a harmonic mean of the sequence, i.e., $\{\tilde{\alpha}_{t-1}\}$, and an exponential smoothing is applied on top of the simple harmonic mean.

**Having an upper-bound stabilizes the algorithm from stochastic optimization perspective**. For example, when the training error of the randomly selected mini-batch at $w_t$ is drastically reduced or approaching zero, the one-step Newton solution in (9) could be very large, i.e. $\tilde{\alpha}_{t-1} \gg 0$, which could be too aggressive to other mini-batch and hence Problem (1) prescribed by the batch. **On the other hand, making the upper-bound adaptive allows the algorithm to adapt to the local geometry and avoid restrictions** on using a large step-size when the algorithm tries to make aggressive progress with respect to Problem (1). With the adaptive upper-bound being derived by an **exponential smoothing of the harmonic mean, the step-size is determined by emphasizing the current estimate of local geometry while taking into account the history of the estimates**. The exponential smoothing further stabilizes the algorithm by balancing the trade-off of being locally focused (with respect to $f_{S_t}$) and globally focused (with respect to $P$).

It is worthwhile to mention that **Algorithm 2 does not require computing extra gradient of $f_{S_t}$ with respect to $w$ if compared with $SARAH$ and $SARAH+$**. At each inner iteration, $t \geq 1$, Algorithm 2 computes $\nabla f_{S_t}(w_{t-1} - \alpha v_{t-1})$ with $\alpha = 0$ just as $SARAH$ and $SARAH+$ would compute $\nabla f_{S_t}(w_{t-1})$, and the only difference is that $\alpha$ is specified as a variable in Pytorch. After the adaptive step-size $\alpha_{t-1}$ is determined (Line 17), Algorithm 2 computes $\nabla f_{S_t}(w_{t-1} - \alpha_{t-1} v_{t-1})$ just as $SARAH$ and $SARAH+$ would compute $\nabla f_{S_t}(w_t)$.

In Section 4 of the main paper, we discussed the sensitivity of Algorithm 2 on the choice of $\gamma$. Here, we present the full results (on 10 chosen datasets for both $\ell^2$-regularized and non-regularized cases) in Figures 8, 9, 10, and 11. Note that, in this experiment, we chose $\gamma \in \{\frac{1}{64}, \frac{1}{32}, \frac{1}{16}, \frac{1}{8}, \frac{1}{4}, \frac{1}{2}\}$, and for each $\gamma$, dataset and case, we used 10 distinct random seeds and ran each experiment for 20 effective passes.

#### B.2.2 Other Algorithms

In our numerical experiment, we compared the performance of **TUNE-FREE** $AI\text{-}SARAH$ (Algorithm 2) with that of 5 **FINE-TUNED** state-of-the-art (stochastic variance reduced or adaptive) first-order methods: $SARAH$, $SARAH+$, $SVRG$, $ADAM$ and $SGD$ with Momentum ($SGD$ w/m). These algorithms were implemented in Pytorch, where $ADAM$ and $SGD$ w/m are built-in optimizers of Pytorch.

---

[6]See code at `https://github.com/shizheng-rlfresh/ai_sarah`.

[7]For detailed description of the automatic differentiation engine in Pytorch, please see `https://pytorch.org/tutorials/beginner/blitz/autograd_tutorial.html`.

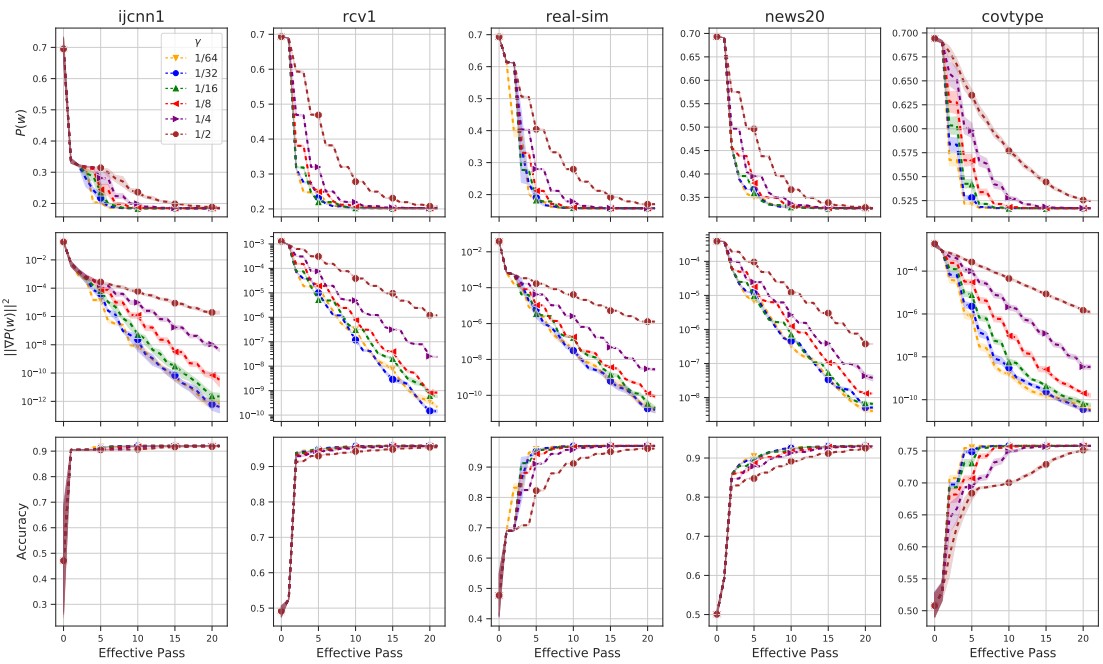

Figure 8: $\ell^2$-regularized case *ijcnn1, rcv1, real-sim, news20* and *covtype* with $\gamma \in \{\frac{1}{64}, \frac{1}{32}, \frac{1}{16}, \frac{1}{8}, \frac{1}{4}, \frac{1}{2}\}$: evolution of $P(w)$ (top row) and $\|\nabla P(w)\|^2$ (middle row) and running maximum of testing accuracy (bottom row).

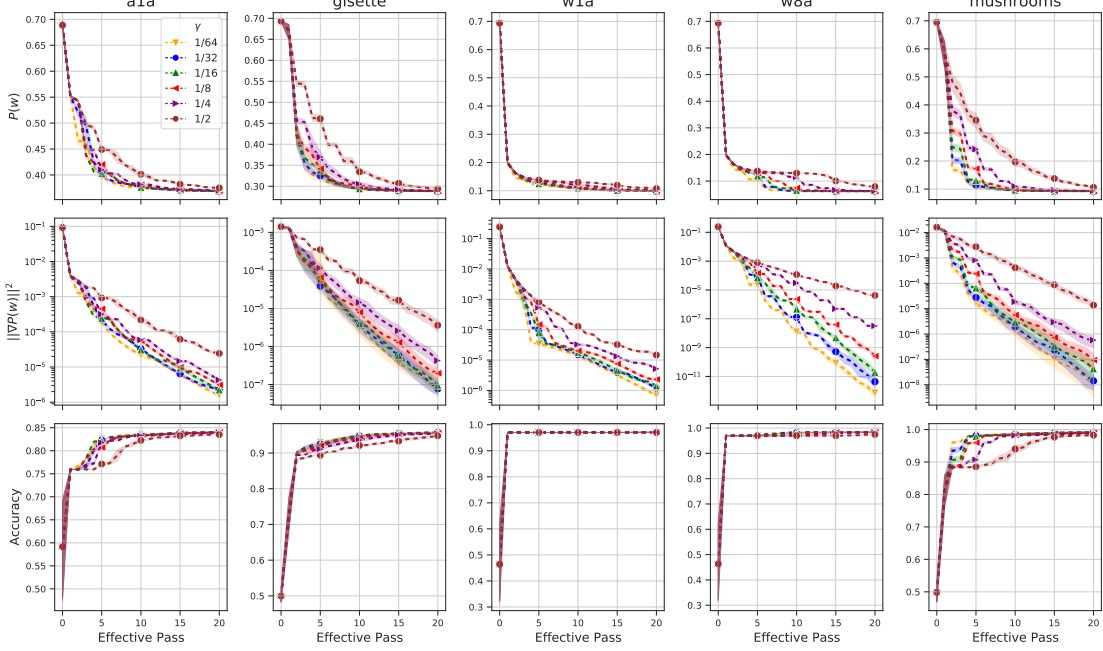

Figure 9: $\ell^2$-regularized case of *a1a, gisette, w1a, w8a* and *mushrooms* with $\gamma \in \{\frac{1}{64}, \frac{1}{32}, \frac{1}{16}, \frac{1}{8}, \frac{1}{4}, \frac{1}{2}\}$: evolution of $P(w)$ (top row) and $\|\nabla P(w)\|^2$ (middle row) and running maximum of testing accuracy (bottom row).

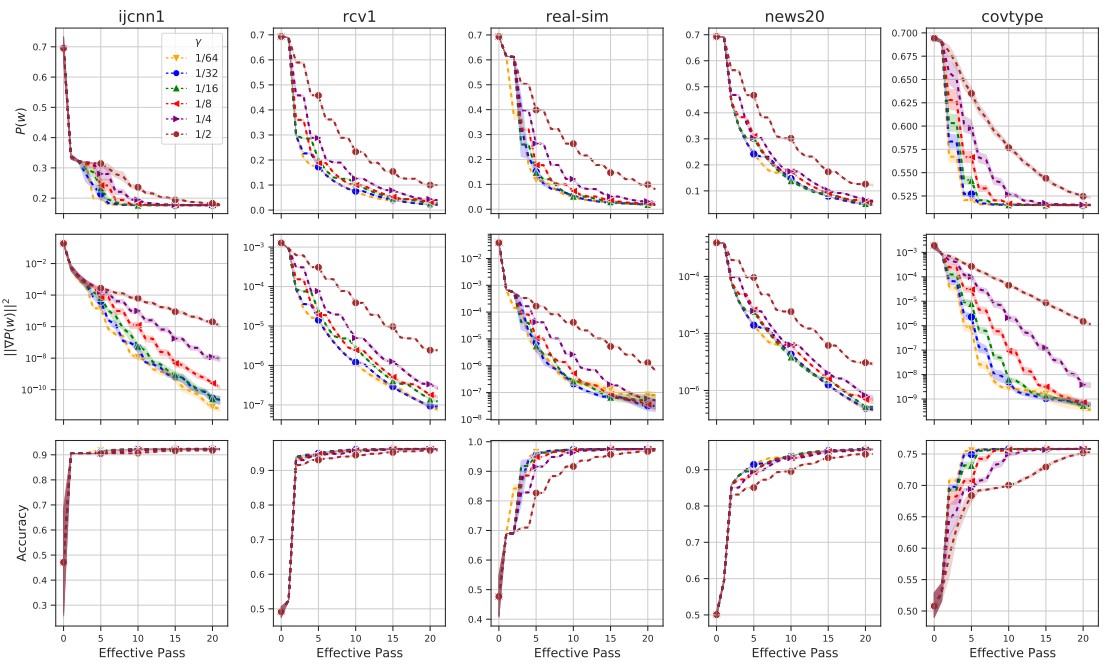

Figure 10: Non-regularized case *ijcnn1, rcv1, real-sim, news20* and *covtype* with $\gamma \in \{\frac{1}{64}, \frac{1}{32}, \frac{1}{16}, \frac{1}{8}, \frac{1}{4}, \frac{1}{2}\}$: evolution of $P(w)$ (top row) and $\|\nabla P(w)\|^2$ (middle row) and running maximum of testing accuracy (bottom row).

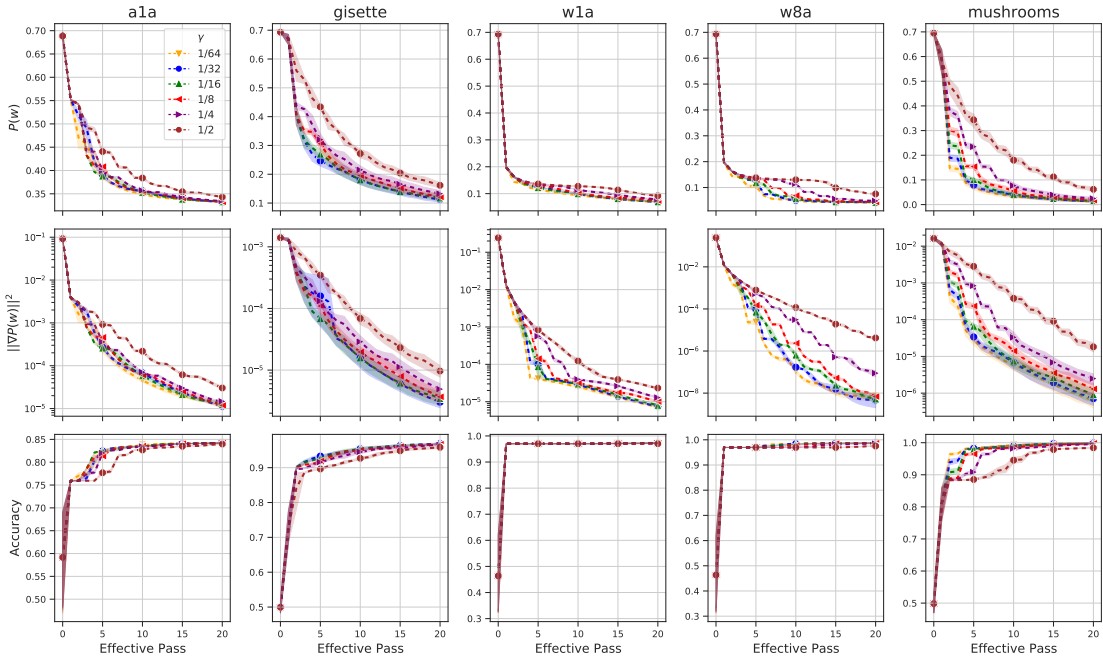

Figure 11: Non-regularized case *a1a, gisette, w1a, w8a* and *mushrooms* with $\gamma \in \{\frac{1}{64}, \frac{1}{32}, \frac{1}{16}, \frac{1}{8}, \frac{1}{4}, \frac{1}{2}\}$: evolution of $P(w)$ (top row) and $\|\nabla P(w)\|^2$ (middle row) and running maximum of testing accuracy (bottom row).

Table 3: Tuning Plan - Choice of Hyper-parameters.

| Method | # Configuration | Step-Size | Schedule (%)[1] | Inner Loop Size (# Effective Pass) | Early Stopping ($\gamma$) |
|---|---|---|---|---|---|
| $SARAH$ | 160 | $\{0.1, 0.2, ..., 1\}/L$ | n/a | $\{0.5, 0.6, ..., 2\}$ | n/a |
| $SARAH+$ | 50 | $\{0.1, 0.2, ..., 1\}/L$ | n/a | n/a | $1/\{2, 4, 8, 16, 32\}$ |
| $SVRG$ | 160 | $\{0.1, 0.2, ..., 1\}/L$ | n/a | $\{0.5, 0.6, ..., 2\}$ | n/a |
| $ADAM$[2] | 300 | $[10^{-3}, 10]$ | $\{0, 1, 5, 10, 15\}$ | n/a | n/a |
| $SGD\ w/m$[3] | 300 | $[10^{-3}, 10]$ | $\{0, 1, 5, 10, 15\}$ | n/a | n/a |

[1] Step-size is scheduled to decrease by $X\%$ every effective pass over the training samples.
[2] $\beta_1 = 0.9, \beta_2 = 0.999$.
[3] $\beta = 0.9$.

Table 4: Running Budget (# Effective Pass).

| Dataset | Regularized | Non-regularized |
|---|---|---|
| $ijcnn1$ | 20 | 20 |
| $rcv1$ | 30 | 40 |
| $news20$ | 40 | 50 |
| $covtype$ | 20 | 20 |
| $real\text{-}sim$ | 20 | 30 |
| $a1a$ | 30 | 40 |
| $gisette$ | 30 | 40 |
| $w1a$ | 40 | 50 |
| $w8a$ | 30 | 40 |
| $mushrooms$ | 30 | 40 |

**Hyper-parameter tuning.** For $ADAM$ and $SGD$ w/m, we selected 60 different values of the (initial) step-size on the interval $[10^{-3}, 10]$ and 5 different schedules to decrease the step-size after every effective pass on the training samples; for $SARAH$ and $SVRG$, we selected 10 different values of the (constant) step-size and 16 different values of the inner loop size; for $SARAH+$, the values of step-size were selected in the same way as that of $SARAH$ and $SVRG$. In addition, we chose 5 different values of the inner loop early stopping parameter. Table 3 presents the detailed tuning plan for these algorithms.

*Selection criteria:*

We defined the best hyper-parameters as the ones yielding the minimum ending value of the loss function, where the running budget is presented in Table 4. Specifically, the criteria are: (1) filtering out the ones exhibited a "spike" of the loss function, i.e., the initial value of the loss function is surpassed at any point within the budget; (2) selecting the ones achieved the minimum ending value of the loss function.

*Hightlights of the hyper-parameter search:*

- To take into account the randomness in the performance of these algorithms provided different hyper-parameters, we ran each configuration with 5 distinct random seeds. **The total number of runs for each dataset and case is** $4,850$**.**
- Tables 5 and 6 present the best hyper-parameters selected from the candidates for the regularized and non-regularized cases.
- Figures 12, 13, 14 and 15 show the performance of different hyper-parameters for all tuned algorithms; it is clearly that, **the performance is highly dependent on the choices of hyper-parameter for $SARAH$, $SARAH+$, and $SVRG$**. And, **the performance of $ADAM$ and $SGD$ w/m are very SENSITIVE to the choices of hyper-parameter**.

**Global Lipschitz smoothness of** $P(w)$**.** Tuning the (constant) step-size of $SARAH$, $SARAH+$ and $SVRG$ requires the parameter of (global) Lipschitz smoothness of $P(w)$, denoted the (global) Lipschitz constant $L$,

Table 5: Fine-tuned Hyper-parameters - $\ell^2$-regularized Case.

| Dataset | ADAM $(\alpha_0, x\%)$ | SGD w/m $(\alpha_0, x\%)$ | SARAH $(\alpha, m)$ | SARAH+ $(\alpha, \gamma)$ | SVRG $(\alpha, m)$ |
|---|---|---|---|---|---|
| ijcnn1 | (0.07, 15%) | (0.4, 15%) | (3.153, 1015) | (3.503, 1/32) | (3.503, 1562) |
| rcv1 | (0.016, 10%) | (4.857, 10%) | (3.924, 600) | (3.924, 1/32) | (3.924, 632) |
| news20 | (0.028, 15%) | (6.142, 10%) | (3.786, 468) | (3.786, 1/32) | (3.786, 468) |
| covtype | (0.07, 15%) | (0.4, 15%) | (2.447, 13616) | (2.447, 1/32) | (2.447, 13616) |
| real-sim | (0.16, 15%) | (7.428, 15%) | (3.165, 762) | (3.957, 1/32) | (3.957, 1694) |
| a1a | (0.7, 15%) | (4.214, 15%) | (2.758, 50) | (2.758, 1/32) | (2.758, 50) |
| gisette | (0.028, 15%) | (8.714, 10%) | (2.320, 186) | (2.320, 1/16) | (2.320, 186) |
| w1a | (0.1, 10%) | (3.571, 10%) | (3.646, 60) | (3.646, 1/32) | (3.646, 76) |
| w8a | (0.034, 15%) | (2.285, 15%) | (2.187, 543) | (3.645, 1/32) | (3.645, 1554) |
| mushrooms | (0.220, 15%) | (3.571, 0%) | (2.682, 190) | (2.682, 1/32) | (2.682, 190) |

Table 6: Fine-tuned Hyper-parameters - Non-regularized Case.

| Dataset | ADAM $(\alpha_0, x\%)$ | SGD w/m $(\alpha_0, x\%)$ | SARAH $(\alpha, m)$ | SARAH+ $(\alpha, \gamma)$ | SVRG $(\alpha, m)$ |
|---|---|---|---|---|---|
| ijcnn1 | (0.1, 15%) | (0.58, 15%) | (3.153, 1015) | (3.503, 1/32) | (3.503, 1562) |
| rcv1 | (5.5, 10%) | (10.0, 0%) | (3.925, 632) | (3.925, 1/32) | (3.925, 632) |
| news20 | (1.642, 10%) | (10.0, 0%) | (3.787, 468) | (3.787, 1/32) | (3.787, 468) |
| covtype | (0.16, 15%) | (2.2857, 15%) | (2.447, 13616) | (2.447, 1/32) | (2.447, 13616) |
| real-sim | (2.928, 15%) | (10.0, 0%) | (3.957, 1609) | (3.957, 1/16) | (3.957, 1694) |
| a1a | (1.642, 15%) | (6.785, 1%) | (2.763, 50) | (2.763, 1/32) | (2.763, 50) |
| gisette | (2.285, 1%) | (10.0, 0%) | (2.321, 186) | (2.321, 1/32) | (2.321, 186) |
| w1a | (8.714, 10%) | (10.0, 0%) | (3.652, 76) | (3.652, 1/32) | (3.652, 76) |
| w8a | (0.16, 10%) | (10.0, 5%) | (2.552, 543) | (3.645, 1/32) | (3.645, 1554) |
| mushrooms | (10.0, 0%) | (10.0, 0%) | (2.683, 190) | (2.683, 1/32) | (2.683, 190) |

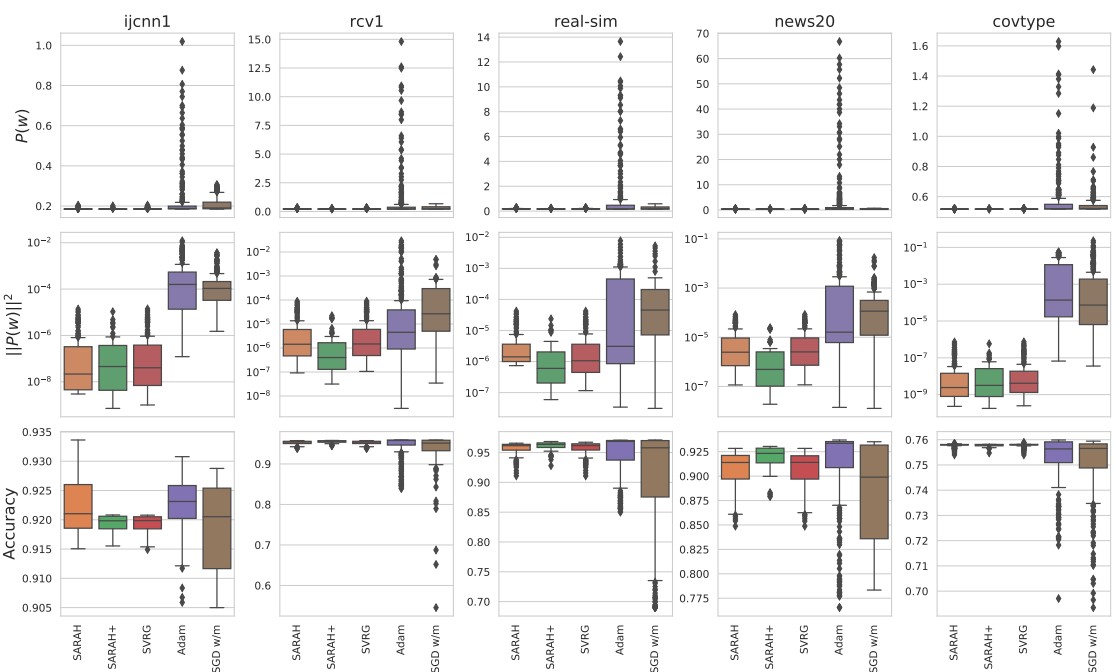

Figure 12: Ending loss (top row), ending squared norm of full gradient (middle row), maximum testing accuracy (bottom row) of different hyper-paramters and algorithms for the $\ell^2$**-regularized case** on *ijcnn1, rcv1, real-sim, news20* and *covtype* datasets.

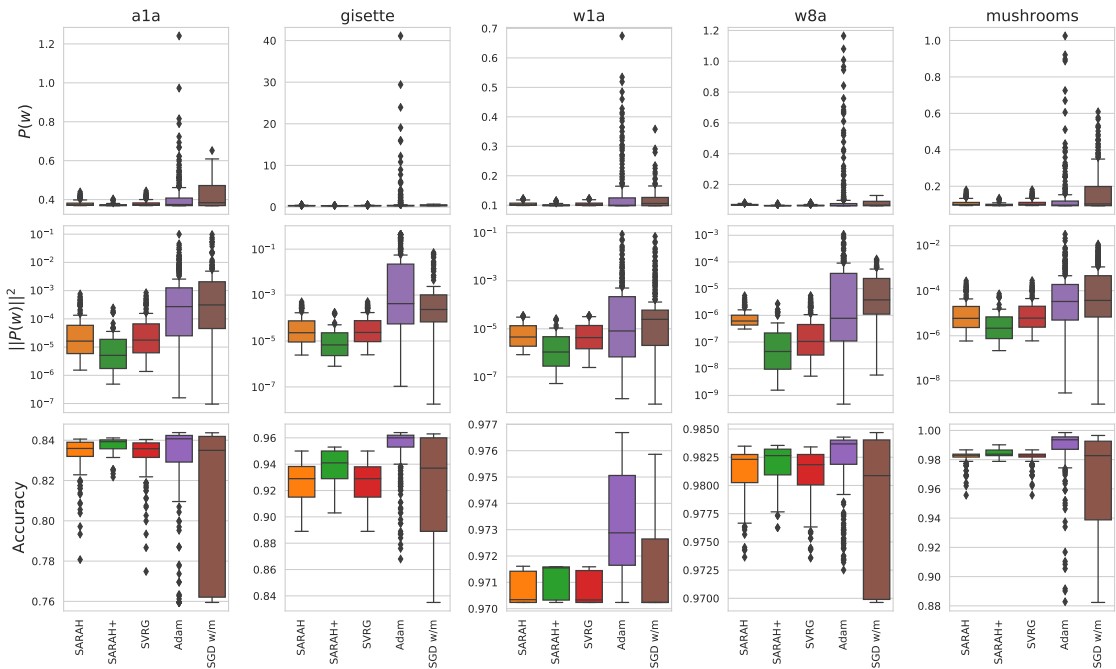

Figure 13: Ending loss (top row), ending squared norm of full gradient (middle row), maximum testing accuracy (bottom row) of different hyper-paramters and algorithms for the $\ell^2$**-regularized case** on *a1a, gisette, w1a, w8a* and *mushrooms* datasets.

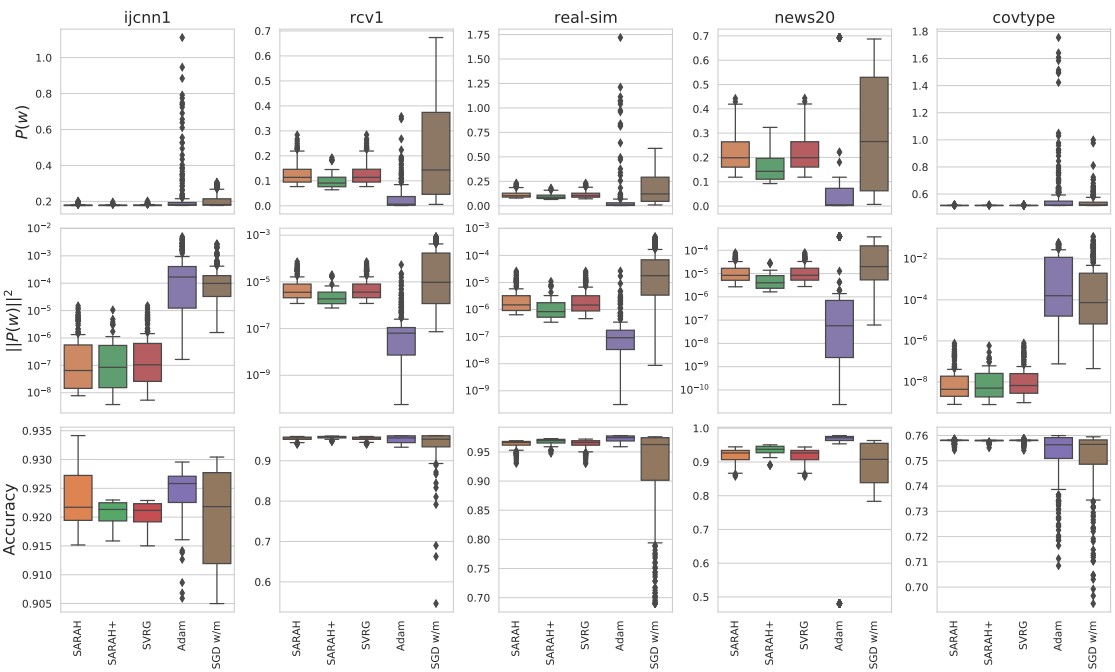

Figure 14: Ending loss (top row), ending squared norm of full gradient (middle row), maximum testing accuracy (bottom row) of different hyper-paramters and algorithms for the **non-regularized case** on *ijcnn1, rcv1, real-sim, news20* and *covtype* datasets.

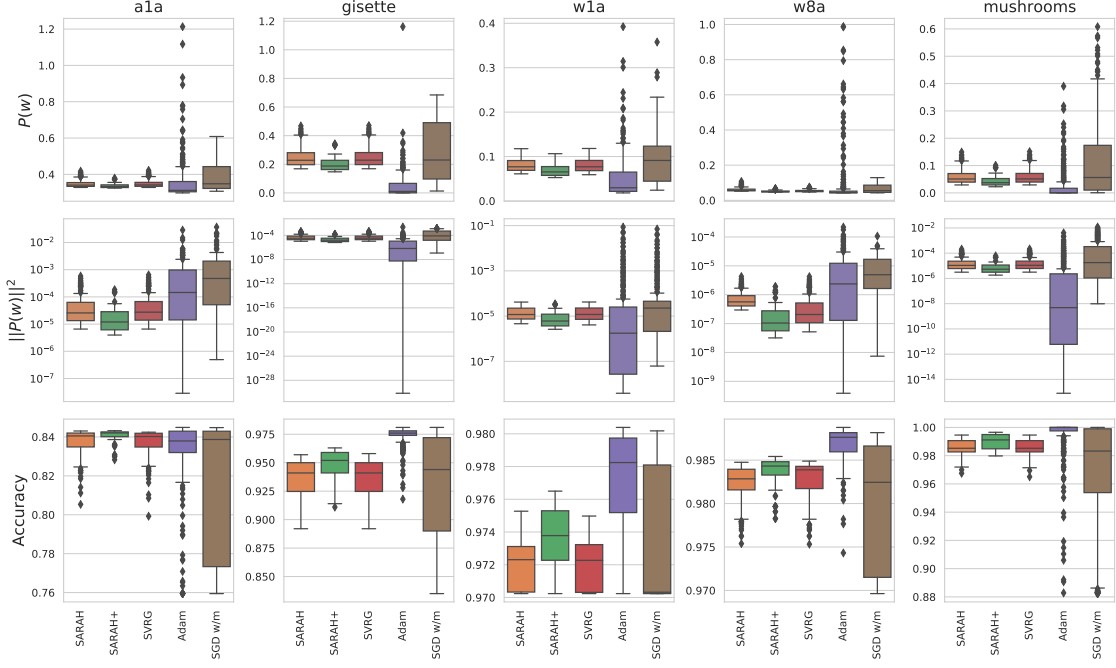

Figure 15: Ending loss (top row), ending squared norm of full gradient (middle row), maximum testing accuracy (bottom row) of different hyper-paramters and algorithms for the **non-regularized case** on *a1a, gisette, w1a, w8a* and *mushrooms* datasets.

Table 7: Global Lipschitz Constant $L$

| Dataset | Regularized | Non-regularized |
|---|---|---|
| *ijcnn1* | 0.285408 | 0.285388 |
| *rcv1* | 0.254812 | 0.254763 |
| *news20* | 0.264119 | 0.264052 |
| *covtype* | 0.408527 | 0.408525 |
| *real-sim* | 0.252693 | 0.252675 |
| *a1a* | 0.362456 | 0.361833 |
| *gisette* | 0.430994 | 0.430827 |
| *w1a* | 0.274215 | 0.273811 |
| *w8a* | 0.274301 | 0.274281 |
| *mushrooms* | 0.372816 | 0.372652 |

and it can be computed as, given (16) and (17),

$$L = \frac{1}{4}\lambda_{max}\left(\frac{1}{n}\sum_{i=1}^{n} x_i x_i^T\right) + \lambda,$$

where $\lambda_{max}(A)$ denotes the largest eigenvalue of $A$ and $\lambda$ is the penalty term of the $\ell^2$-regularization in (16). Table 7 shows the values of $L$ for the regularized and non-regularized cases on the chosen datasets.

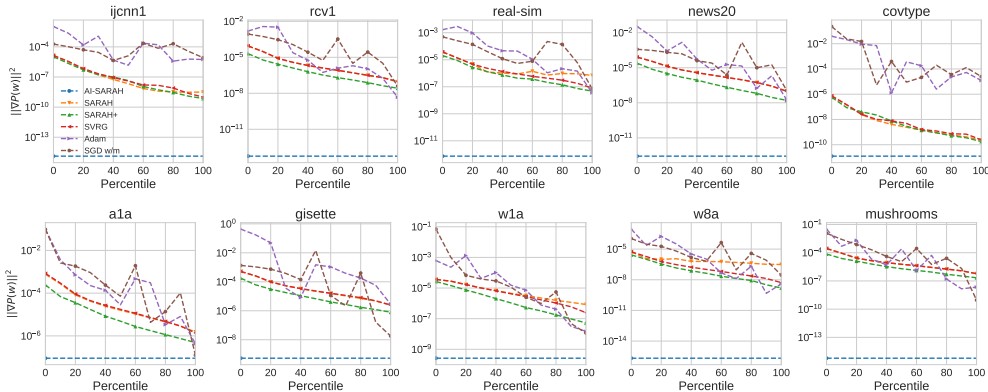

Figure 16: Average ending $\|\nabla P(w)\|^2$ for $\ell^2$-regularized case - *AI-SARAH* vs. Other Algorithms: *AI-SARAH* *is shown as the horizontal lines; for each of the other algorithms, the average ending* $\|\nabla P(w)\|^2$ *from different configurations of hyper-parameters are indexed from* 0 *percentile (the worst choice) to* 100 *percentile (the best choice); see Section B.2.2 for details of the selection criteria.*

## B.3 Extended Results of Experiment

In Section 5, we compared tune-free & fully adaptive *AI-SARAH* (Algorithm 2) with fine-tuned *SARAH*, *SARAH+*, *SVRG*, *ADAM* and *SGD* w/m. In this section, we present the extended results of our empirical study on the performance of *AI-SARAH*. For the experiments, we used NVIDIA V100 GPUs.

Figures 16 and 17 compare the average ending $\|\nabla P(w)\|^2$ achieved by *AI-SARAH* with the other algorithms, configured with all candidate hyper-parameters.

It is clear that,

- without tuning, *AI-SARAH* achieves the best convergence (to a stationary point) in practice on most of the datasets for both cases;
- while fine-tuned *ADAM* achieves a better result for the non-regularized case on *a1a, gisette, w1a* and *mushrooms*, *AI-SARAH* outperforms *ADAM* for at least 80% (*a1a*), 55% (*gisette*), 50% (*w1a*), and 50% (*mushrooms*) of all candidate hyper-parameters.

Figure 18 shows the results of the non-regularized case for *ijcnn1, rcv1, real-sim, news20* and *covtype* datasets. Figures 19 and 20 present the results of the $\ell^2$-regularized case and non-regularized case respectively on *a1a, gisette, w1a, w8a* and *mushrooms* datasets. For completeness of presentation, we present the evolution of *AI-SARAH*'s step-size and upper-bound on *a1a, gisette, w1a, w8a* and *mushrooms* datasets in Figures 21 and 22. Consistent with the results shown in Section 5 of the main paper, *AI-SARAH* delivers a competitive performance in practice.

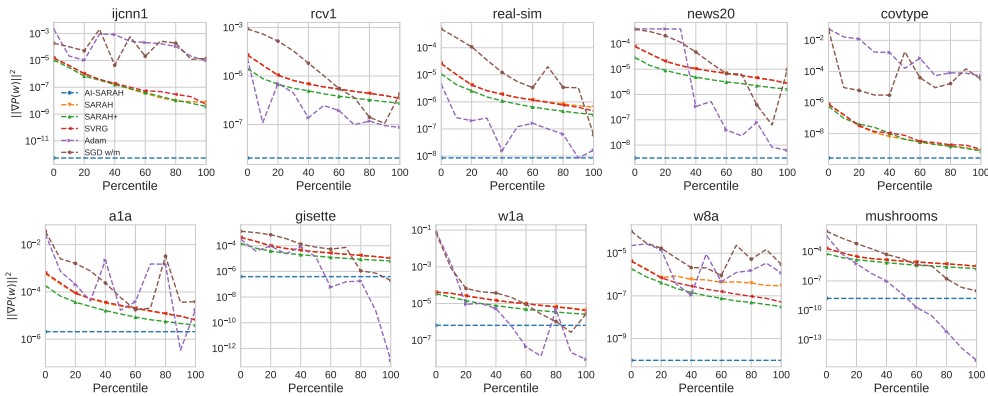

Figure 17: Average ending $\|\nabla P(w)\|^2$ for non-regularized case - *AI-SARAH* vs. Other Algorithms.

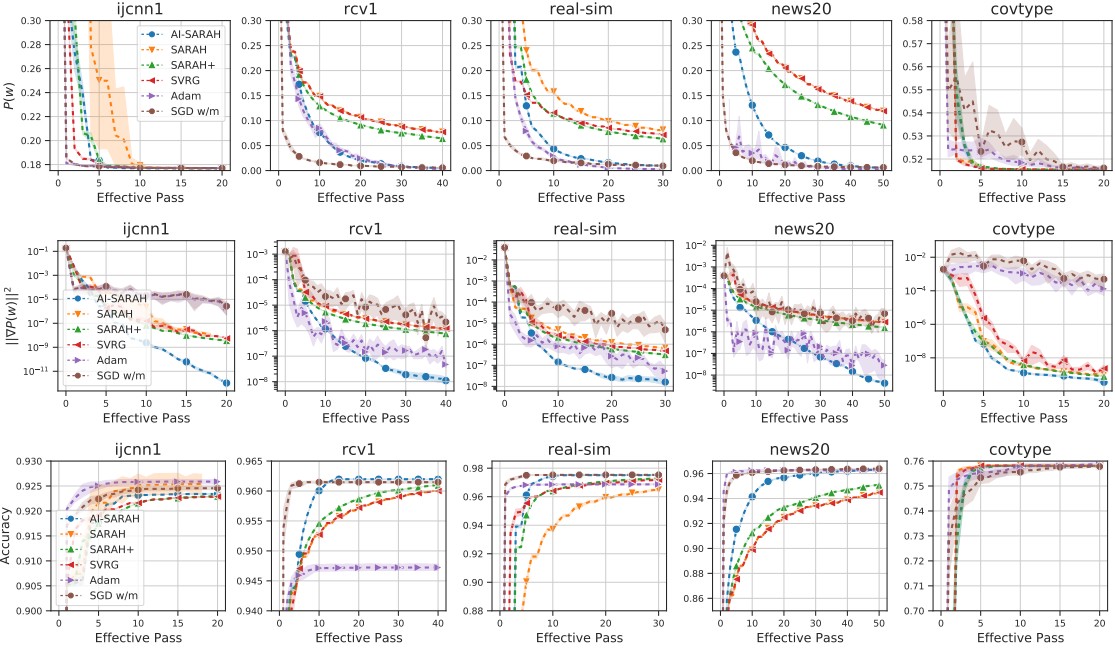

Figure 18: Non-regularized case: evolution of $P(w)$ (top row), $\|\nabla P(w)\|^2$ (middle row), and running maximum of testing accuracy (bottom row).

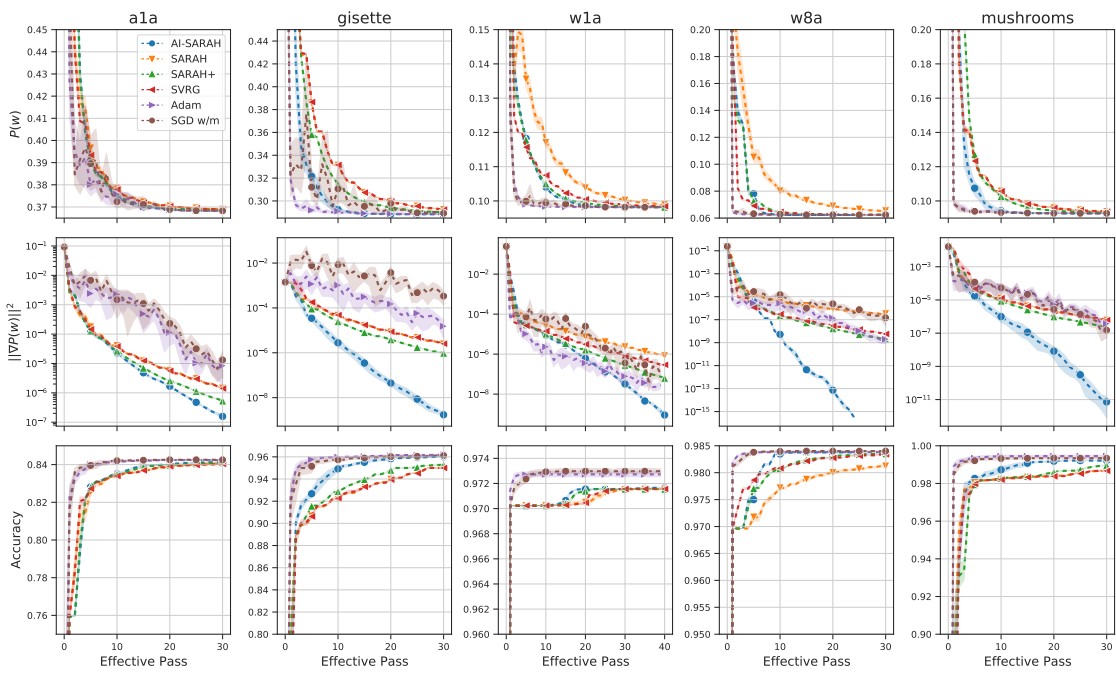

Figure 19: $\ell^2$-regularized case: evolution of $P(w)$ (top row), $\|\nabla P(w)\|^2$ (middle row), and running maximum of testing accuracy (bottom row).

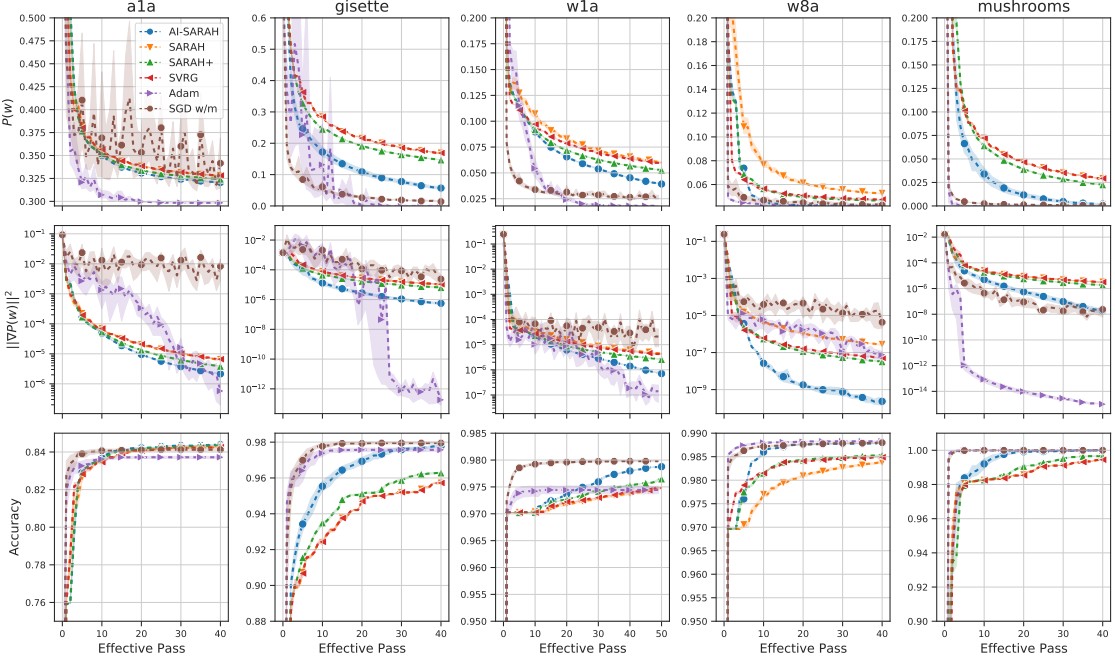

Figure 20: Non-regularized case: evolution of $P(w)$ (top row), $\|\nabla P(w)\|^2$ (middle row), and running maximum of testing accuracy (bottom row).

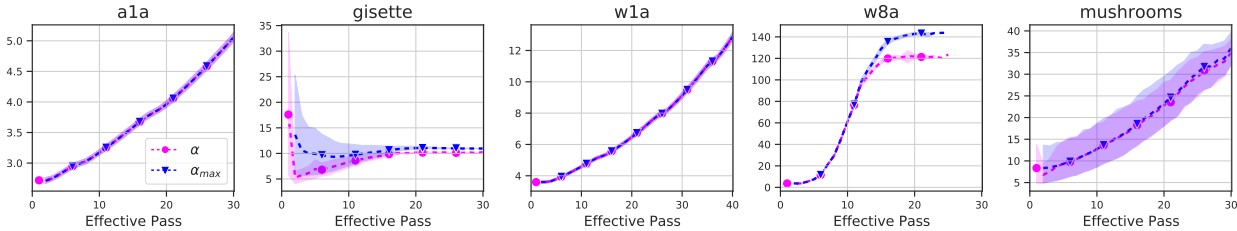

Figure 21: $\ell^2$-regularized case: evolution of *AI-SARAH*'s step-size $\alpha$ and upper-bound $\alpha_{max}$.

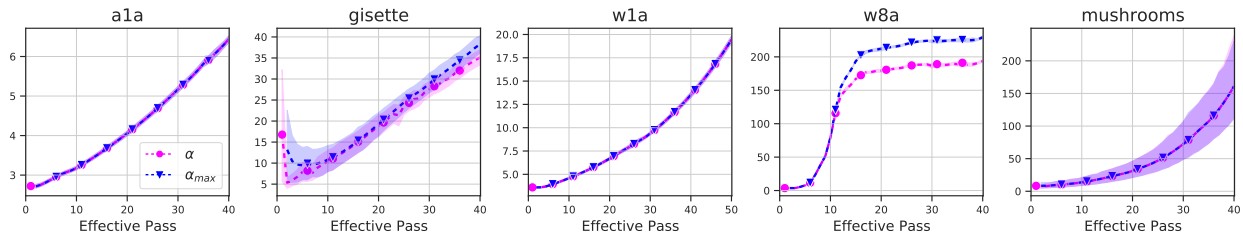

Figure 22: Non-regularized case: evolution of *AI-SARAH*'s step-size $\alpha$ and upper-bound $\alpha_{max}$.

