# OpenReview forum: "AI-SARAH: Adaptive and Implicit Stochastic Recursive Gradient Methods"
_TMLR — Accepted by TMLR_

### Review · Reviewer_PreZ · 2022-10-17

**Summary Of Contributions:**

This paper aims to make SARAH more efficient in practical problems. The authors propose a variant of SARAH, that is tune-free and adaptive. The authors present the convergence of the variant and discuss the theoretical performance under different sampling styles. Experimental results are solid and well discussed. I deem the paper is beyond the acceptance bar.



**Audience:**

Yes

**Broader Impact Concerns:**

I do not find any ethical concerns.

**Claims And Evidence:**

Yes

**Requested Changes:**

1.Add a large-scale  test.

2.Add more explantions about the sub-problem approximation.

**Strengths And Weaknesses:**

Strength：1. The paper is well-motivated. The authors present an example demonstrating that we need to use the local Lipschitz smoothness property rather than the global one.

2. The advantage of the proposed algorithm is obvious. The algorithm is adaptive and needs a few tunings. The presented numerical results show that the algorithm performs relatively well: the algorithm is faster than others with respect to both iterations and time.

3. The algorithm is well discussed, including the convergence, results under different samplings, subproblems, and stepsize. Although the authors do not present the theoretical accelerations, existing convergence results are satisfying because they focus on developing a practical efficient algorithm. Indeed, I believe it is very difficult to prove the theoretical acceleration even with extra assumptions.

Weakness:
1. Sub-problem (10) in Algorithm 2 is vital for the algorithm. I appreciate that the authors present many details about the sub-problem. The authors mentioned that automatic differentiation could directly solve the sub-problem in Pytorch. To eliminate extra gradient calculations, the authors approximate (6) at one-step Newton after $\alpha=0$. But how to guarantee that the approximation is acceptable?

2. I appreciate that the authors present many numerics. But the tests are really small. A large-scale test is optimal.

---

> ### Author Response · Authors · 2022-12-11
> **Thank you for your review**
>
> We would like to thank the reviewer for carefully reading our paper, and we are happy that he liked the main idea of the paper - to develop an adaptive algorithm. We also agree with his observation that it would be super hard to prove theoretical acceleration for the practical algorithm.
>
> Let us address the weaknesses and requested changes below.

---

> > ### Author Response · Authors · 2022-12-11
> > **Sub-problem  Algorithm 2, Line 10**
> >
> > We have added more details into paragraph *AI-SARAH sub-problem* in Section 4.1. Also check our response to reviewer Eu4H.
> > We also added some more explanations in Section 4.2 why one Newton-step could be a good choice

---

> > > ### Author Response · Authors · 2022-12-11
> > > **A large-scale test**
> > >
> > > News20 has dimension $d = 1,355,191$ and $n=14,997$ samples.
> > >
> > > Another one that has more samples is covtype with $n=435,759$ (however, only 54 features).
> > >
> > > You are correct that one could take maybe
> > >
> > > KDD2021 algebra https://www.csie.ntu.edu.tw/~cjlin/libsvmtools/datasets/binary.html#kdd2010%20(algebra
> > >
> > > with 8.5M features (which has 6x more features when compared to News20). The main reason why we haven’t run it was the extra cost for the other benchmarks (recall, we run almost 5k settings for other algorithms). But if you think that adding this (or feel free to suggest another dataset) would bring some new insides, we would be happy to run it and include it in the paper. But please suggest the dataset we should run it on, as it is quite expensive to fine-tune other benchmarks.

---

### Review · Reviewer_Eu4H · 2022-10-19

**Summary Of Contributions:**

This paper proposed a practical variant of SARAH for strongly convex problems, which adjusts step-sizes based on local geometry with little extra cost compared with the original SARAH. The proposed method is tune-free, and numerical experiments demonstrate the efficiency comparing with other state-of-the-art methods with fine-tuned hyper-parameters.

**Audience:**

Yes

**Broader Impact Concerns:**

This is a theoretical paper and I think there is no ethical concerns.

**Claims And Evidence:**

Yes

**Requested Changes:**

1. I would give a positive suggestion if the authors could provide more theoretical guarantees of (6). For example, it would be appreciated if the authros could prove the convergence of a variant of Algorithm 1, which only replaces lines 7 and 15 by (6).

2. Other comments:

1). The authors claimed that "At every iteration, instead of incurring expensive costs on computing the parameters of local Lipshitz smoothness for all f i in Algorithm 1, Algorithm 2 estimates the local smoothness by approximately solving the sub-problem for only one $f_i$". However, I think it is unfair to compare a theoretically correct algorithm to an empirical one. If we ignore the theoritical guarantees, it would be more nature to compute only one $L_i^t$ on lines 7 and 15 in Algorithm 1.

2). I am not convinced that (6) approximates the local geometry. I think it searches the step size $\alpha$ to minimize $\|v_t\|$, and accordingly, to minimize $\|\nabla f(w_t)\|$ finally.

3). it would be appreciated if the authros could provide more experimental resutls on the case of $\beta=0$, which could remove the other tricks and only verify the efficiency of (6).

**Strengths And Weaknesses:**

Strengths:

1. The authors proposed a new strategy to adjust step-sizes based on local geometry, especially for the methods not necessarily moving along the descent direction.

2. The efficiency is verified empirically by extensive numerical experiments on various datasets.

Weaknesses:

1. The authors only prove the convergence for the theoretical-AI-SARAH, which is not practical. No theoretical guarantees is provided for the more interesting AI-SARAH.

2. The proof of Theorem 3.1 is almost the same with those in the original SARAH paper (Nguyen et al. 2017).

---

> ### Author Response · Authors · 2022-12-05
> **Thank you**
>
> We thank the reviewer Eu4H for carefully reading our paper.
>  Let us address the weaknesses that you have pointed out in separate threads.

---

> > ### Author Response · Authors · 2022-12-05
> > **Issue: No theoretical guarantees is provided for the more interesting AI-SARAH.**
> >
> > Indeed, we have not provided the theoretical guarantees for the practical version of the algorithm, not because we have not tried, but simply because we believe it is very hard to get some nice theoretical guarantees for the practical algorithm.
> >
> >
> > Let us remark that this is not something completely new as a similar approach is made in many papers, where a theoretical algorithm is proposed and analyzed and when it goes to numerical experiments, people make multiple tricks to make it work. For example, some algorithms require step-size to be in order $O(\epsilon)$ to get the convergence rate, but when implemented, they would choose the step-size not following the theory, but most often using some heuristics (quite often only mentioned in numerical experiments or appendix).
> >
> > In our paper, we approached this problem slightly differently. We have been quite open about this, and we clearly stated that AI-SARAH is a practical algorithm inspired by the theoretical one. One can hence see AI-SARAH as a heuristic version of the theoretical algorithm.

---

> > > ### Author Response · Authors · 2022-12-05
> > > **The proof of Theorem 3.1 is almost the same with those in the original SARAH paper (Nguyen et al. 2017).**
> > >
> > > Indeed, our proofs are motivated by (Nguyen et al. 2017); however, we had to make some refinements as we needed to be sure that the step size (which can be larger than the one in the original SARAH) would not make the lemmas and theorems invalid. We have added a sentence in Section A.1 stating that we are extending the Lemmas from (Nguyen et al. 2017).

---

> > > > ### Author Response · Authors · 2022-12-05
> > > > **Requested Changes**
> > > >
> > > > Unfortunately, we have not been able to derive such a theoretical guarantee where we would assume that we solve (6) (which can be a non-convex problem even if the f_i are convex) and hence we introduce  Lines 7 and 15 :(
> > > > Other comments:
> > > >
> > > > 1. It turns out that in order to make some steps in the proof, the step-size has to be chosen in such a way that all L_i^t are correct. Therefore computing only one L_i^t would, in theory, not lead to a convergent algorithm, but it would definitely be a good heuristic.
> > > >
> > > >
> > > > 2. Note that $\|v_t\|$ has two parts. The fixed vector $v_{t-1}$ (which is also the direction we are moving from w_{t-1} to w_{t}) and the function $f_i$.
> > > > Hence, we can learn something about the change of the gradient of $f_i$ in the direction of update $v_{t-}$. See more details in Section 4.1, and please let us know if more explanation would be needed.
> > > > If we assume that the Hessian of the $f_i$ is not changing much, then we can approximate $\nabla f_i(w - \alpha v) - \nabla f_i(w)$ by
> > > > $ \nabla^2 f_i(w) (-\alpha v) $
> > > > and then the optimal solution of $\min_{\alpha} \xi(\alpha)$ would be
> > > > $\alpha ^* = (v^T \nabla^2 f_i  v) / (  v^T \nabla^2 f_i \nabla^2 f_i v )$
> > > > which is nothing else than roughly speaking estimating a curvature of $f_i$ in the direction of update $v$.
> > > > Note that as written in Section 4.1 for a quadratic function $f_i$, the Hessian is constant, and we are getting an estimate of its largest eigenvalue.
> > > >
> > > >
> > > > 3.  We initially did not have $\beta$ hence $\beta$ was 0. And this algorithm is not working simply because the step size would completely ignore other functions. Recall the theoretical SARAH defines step size that depends on $L$ and that depends on all functions $f_i$. So basically, picking one function that is “too flat” would make a huge update to $w$. Hence, in some sense, we are using an exponential moving average of local estimates of $L_i$, which is a classical trick in many optimization algorithms for ML (e.g., Adam).

---

> > > ### Comment · Reviewer_Eu4H · 2022-12-13
> > > **resposne**
> > >
> > > I agree with that

---

### Review · Reviewer_sJon · 2022-11-28

**Summary Of Contributions:**

This paper presents a tune-free variant of the SARAH algorithm for finite-sum optimization. The algorithm chooses a step size based on the local geometry of the problem and can thus adapt to changing flatness and sharpness of the loss landscape. The authors present a theoretical version of the algorithm and give a theoretical analysis that improves SARAH. However, the theoretical version is extremely expensive as it requires computing the Lipschitzness of the local hessian $tnk$ times in a ball determined by the step size (which itself depends on the local step size), where $n$ is the number of components in the finite sum, and $t,k$ are inner and outer iteration numbers.

To alleviate this, the authors present a lighter version of the algorithm, which only approximately computes this lipschitzness for just one component in each inner and outer iteration. No theoretical guarantees are provided for this lighter version, but it is extensively compared against the other baselines empirically.

**Audience:**

Yes

**Broader Impact Concerns:**

No broad impact concerns.

**Claims And Evidence:**

Yes

**Requested Changes:**

1. It would help to add explicit updates for different sampling strategies in algorithm one and retain the discussion in the following text. Currently, some lines are too abstract to make out what is happening in the algorithm, as the pseudo-code is not self-contained. These are mentioned in the appendix, but combining them in the main paper would be good. Line 13 is confusing because it only works for uniform sampling (again, algorithms 3 and 4 are more explicit and useful). It might also be helpful to highlight in a different color the implicit computations in the theoretical algorithm, which are very expensive, so that the changes made in the practical algorithm are easily comparable.

2. Although two sampling strategies are proposed for algorithm 1, the authors are not explicit about which strategy is used for the guarantee in theorem 3.1. Based on the appendix, the result is for uniform sampling; this should be explicit. Also, it is unclear why the authors are presenting these strategies in the first place. The improvement with importance sampling: going from max to the average highest eigenvalue, is standard and doesn't seem surprising. The experiments seem to use uniform sampling as well.

3. It is unclear why the gradient norm is relevant for regularized logistic regression, which is a strongly convex problem. AI-SARAH is competitive for minimizing function value but only outperforms the baselines for gradient norm minimization. Can the authors comment on why they include the norm minimization experiments? Similarly, the accuracy experiments are not helpful, as the paper is purely about optimization. Were the authors hoping to make some comments about the implicit regularization of their algorithm?

4. It would be interesting to see some experiments which can underline the geometric adaptivity of AI-SARAH. Specifically, a synthetic experiment where the algorithms with any fixed step size perform poorly. I am curious how other adaptive methods, such as ADAM/ADAGRAD/RMSPROP (admittedly not tune-free), perform against AI-SARAH on such a task.


**Strengths And Weaknesses:**

The paper has an extensive empirical evaluation against popular optimization methods. All error bars and the experimental setup are very explicit. The algorithm's theoretical version presents an improvement over the convergence guarantee for SARAH.

The authors don't compare their theoretical guarantee to any other algorithm variance reduced or not. Several tune-free variants of SGD have also emerged in recent years. For non-convex problems, SARAH is optimal, and thus it is reasonable to only compare against it for those problems. However, for strongly convex problems, what's the motivation for not comparing to other algorithms? Are there any relevant lower bounds for finite-sum optimization the authors can discuss?

There are no theoretical guarantees for the practical variant. Based on the approximations the authors suggest in section four, obtaining such a guarantee as a function of perhaps the accuracies for each of these steps seems possible. This might be useful even for a simple problem class like quadratic problems or logistic regression.

Overall, I like the paper, but I believe a better comparison to some existing guarantees and some writing changes, which I suggest below, will improve the submission.

---

> ### Author Response · Authors · 2022-12-14
> **Thank you**
>
> We thank the reviewer sJon for carefully reading our paper. Let us address the weaknesses that you have pointed out in separate threads. Sorry for the late response; the NeurIPS conference cased some delays for us to respond.

---

> > ### Author Response · Authors · 2022-12-17
> > **Experiments which can underline the geometric adaptivity of AI-SARAH (comparison with ADAM/ADAGRAD/RMSPROP)**
> >
> > We tried to highlight the ability of AI-SARAH to adapt to local curvature in Figure 1. We compared it with a fixed step size for SARAH only.
> > In theory, the issue is that one has to choose a fixed step size that is not too large to prevent diverging the classical SARAH algorithm. Note that step-size 16 already led to an initial increase in the objective function.
> > However, AI-SARAH increased step size above 32 after seven epochs as the curvature became flat.
> >
> > ADAGRAD is a bit the opposite, as the step size should never increase (per feature).
> > Adam + RMSPROP are also using a different scaling per feature; hence the effective step size is not unique per feature.
> > Do you suggest we would make maybe a variant of Figure 1 but run Adam/Adagrad and RMSPROP with different step-sizes? Note that Adam and RMSPROP need a scheduler for the learning rate to converge (as they are not variance-reduced methods). Adagrad, on the other hand, is reducing the learning rate to zero; hence a convergence can be achieved.
> >
> > Please advise what plot you would suggest we make, given the issues mentioned above.
> >
> > Thanks

---

> > > ### Author Response · Authors · 2022-12-17
> > > **Theoretical guarantee comparison**
> > >
> > > Thank you for the suggestion. The original SARAH paper and many follow-up papers have included many comparisons to other methods. Hence we believe this is somehow well documented. In this paper, we wanted to show that the adaptive variant is as useful as the original SARAH without requiring an expensive tuning phase.
> > > About the lower bounds - do you have any particular paper(s) in mind that would be good to mention in our setting? Note that we compute full gradient from time to time, so it is not a pure SGD setting.
> > >
> > > [Theoretical comparison with other tune-free variants]
> > > We are aware of a paper that combines SVRG with Adagrad steps https://arxiv.org/abs/2102.09645 that we can mention. Do you have any other suggestions?
> > >
> > > Thanks

---

> > > > ### Author Response · Authors · 2022-12-18
> > > > **Issue related to different sampling strategies**
> > > >
> > > > Thank you for the suggestion. Indeed, we used uniform sampling in experiments but made the theory a bit more general to allow the importance sampling. We also agree that going from uniform to importance sampling is going to change the constant from max to average, which, as you mention, is not so surprising. Let us remark that we spent only 2 lines in the main paper addressing this.
> > > >
> > > > We will add comments to the main paper, as you suggested, to make it less confusing and more self-contained.

---

> > > > > ### Author Response · Authors · 2022-12-21
> > > > > **Gradient norm minimization and accuracy plots**
> > > > >
> > > > > The reason why we plot the size of the gradient is just to show some quantity that converges to 0 as we are approaching the optimal solution. Sometimes people in papers estimate the optimal function value $P^*$ by running the algorithm for "long enough", but we believe that in a strongly convex setting, we can show the size of the gradient instead (which serves up to a constant as upper bound on $P(w_t)-P^*$).
> > > > > Note that by showing only  $P(w_t)$ it is harder to see the difference between various algorithms near the optimal solution.
> > > > >
> > > > > We also included the accuracy plots just for completeness, and we agree that this paper is purely about optimization, but maybe some practitioners would like to see them. We had no intention of showing them related to the implicit regularization of our algorithm.

---

> > > > > > ### Comment · Reviewer_sJon · 2023-01-02
> > > > > > **Makes sense**
> > > > > >
> > > > > > It makes sense; it would be good to include them just for completeness, as the authors mentioned. My original comment highlighted that for strongly convex functions, we expect to get an upper bound on the function sub-optimality (or even iterate distance), which, as you concede, is a stronger guarantee. So it doesn't make sense not to plot them as well. And based on the experiments, there seems to be some difference between the relative behavior while plotting these different quantities. I don't think it is any harder to compare function sub-optimalities. Maybe it makes sense to plot relative function sub-optimality (i.e., normalized by the best function value obtained by newton's method, let's say) if the authors want the plotted quantity to approach zero.

---

> > > > > ### Comment · Reviewer_sJon · 2023-01-02
> > > > > **Thanks**
> > > > >
> > > > > Thanks, that makes sense. In the original submission, I found that the discussion about different sampling strategies did not add much to the table. This is why I think it can be avoided altogether. Given that the authors concede it is not a contribution, why do they want to include this?

---

> > > > ### Comment · Reviewer_sJon · 2023-01-02
> > > > **Theoretical Comparison**
> > > >
> > > > There are a couple of other tune-free methods by some authors of the paper mentioned, such as this one: https://proceedings.mlr.press/v130/loizou21a.html This is just a representative paper, the cited references and follow-up works are also relevant. After reading the submission, it was unclear that the writer is aware of this line of work of tune-free methods and compares against them comprehensively. I believe the point of a good paper with guarantees is not just to provide those guarantees but to put them into context. I agree some of the comparisons might be well documented in older papers, but if one has to go back to reading them, the paper is not self-contained. Regarding lower bounds, there are several lower bounds applicable to the finite sum setting. Perhaps this one might be relevant: https://arxiv.org/abs/1410.0723
> > > >
> > > > Overall, I want the paper to have a well-versed and comprehensive discussion about what is already known and what can be done here. Reading it currently, I don't get that impression.

---

> > > > > ### Author Response · Authors · 2023-02-20
> > > > > **adding a summary of recent development in adaptive step-size**
> > > > >
> > > > > Thank you, we have added details about recent work related to a stochastic polyak step-size and mentioned some of the recent extensions. This can be found in blue in Section 1.2 "Related work".
> > > > >
> > > > > Does this address your concern?

---

> > > ### Comment · Reviewer_sJon · 2023-01-02
> > > **Proposed experiment**
> > >
> > > I agree that figure 1 is trying to highlight the adaptivity of AI-SARAH.
> > >
> > > > Please advise what plot you would suggest we make, given the issues mentioned above.
> > >
> > > I wondered if there is a reasonable experiment where adaptivity is essential for good performance. For e.g., a smaller step size is optimal initially and a larger one later. But I realize this is a more subtle point when comparing against not just SARAH but other adaptive algorithms (which might require a scheduler). Adding the discussion the authors mentioned in the response would be useful. For just comparing against SARAH, I realize now that figure 1 b already does what I wanted.
> > >
> > > > Do you suggest we would make maybe a variant of Figure 1 but run Adam/Adagrad and RMSPROP with different step-sizes?
> > >
> > > This could be useful as well.
> > >
> > > > Note that Adam and RMSPROP need a scheduler for the learning rate to converge (as they are not variance-reduced methods). Adagrad, on the other hand, is reducing the learning rate to zero; hence a convergence can be achieved.
> > >
> > > It would be good to discuss these subtleties of the step-size

---

> > > > ### Author Response · Authors · 2023-02-20
> > > > **Showing slower performance for Adam, Adagrad and RMSProp**
> > > >
> > > > Thank you for your suggestion. We have included Figures 1e) and 1f) to show the comparison of AI-SARAH with the Adam / Adagrad / RMSProp. As expected, the large step-size at the beginning leads to divergence, Adam and RMSProp will also get stuck in a neighborhood of optimal solution due to fixed step-size and because they are not variance reduced.
> > > > Adagrad is basically decreasing step-size. Hence it is eventually converging to the optimal solution, but it is sensitive to step-size selection, as shown in Figure 1f).
> > > >
> > > > Please let us know if you have any further requests.

---

> > ### Comment · Reviewer_sJon · 2023-01-02
> > **My first requested change**
> >
> > Thanks for making the first requested change. I think te readability of the algorithm has certainly improved.

---

### Decision · Action_Editors · 2023-02-22

**Recommendation:** Accept as is

**Comment:**

All reviewers felt the paper was acceptable. Some minor lingering concerns (around motivation, related work, some clarity issues) were addressed in another round of revisions by the authors.

**Audience:**

This is a paper on optimization of functions, exploiting local smoothness. Optimization is a key computational primitive in machine learning and so this paper has broad appeal.

**Claims And Evidence:**

The paper presents a practical version of a theoretically motivated optimization algorithm. Numerical experiments support the claim that the practical version possesses some of the adaptivity that its theoretical counterpart possesses by design.